# ON ROBUST OVERFITTING: ADVERSARIAL TRAINING INDUCED DISTRIBUTION MATTERS

## ABSTRACT

Adversarial training may be regarded as standard training with a modified loss function. But its generalization error appears much larger than standard training under standard loss. This phenomenon, known as robust overfitting, has attracted significant research attention and remains largely as a mystery. In this paper, we first show empirically that robust overfitting correlates with the increasing generalization difficulty of the perturbation-induced distributions along the trajectory of adversarial training (specifically PGD-based adversarial training). We then provide a novel upper bound for generalization error with respect to the perturbation-induced distributions, in which a notion of the perturbation operator, referred to "local dispersion", plays an important role.

## 1 INTRODUCTION

Despite their outstanding performance, deep neural networks (DNNs) are known to be vulnerable to adversarial attacks where a carefully designed perturbation may cause the network to make a wrong prediction (Szegedy et al. (2014), Goodfellow et al. (2015)). Many methods have been proposed to improve the robustness of DNNs against adversarial perturbations (Madry et al. (2019), Zhang et al. (2019), Croce et al. (2020)), among which Projected Gradient Descend based Adversarial Training (PGD-AT) (Madry et al., 2019) is arguably the most effective (Athalye et al. (2018), Dong et al. (2020)). A recent work in Rice et al. (2020) however revealed a surprising phenomenon in PGD-AT: after training, even though the robust error (i.e., error probability in the predicted label for adversarially perturbed instances) is nearly zero on the training set, it may remain very high on the testing set. For example, on the testing set of CIFAR10, the robust error of PGD-AT trained model can be as large as 44.19%. This significantly contrasts the standard training: on CIFAR10, when the standard error (i.e., the error probability in the predicted label for non-perturbed instances) is nearly zero on the training set, its value on the testing set is only about 4%. This unexpected phenomenon arising in PGD-AT is often referred to as robust overfitting.

Since its discovery, robust overfitting has attracted significant research attention. A great deal of research effort has been spent on understanding its cause and devising mitigating techniques. The work of Wu et al. (2020) and Stutz et al. (2021) correlate robust overfitting with sharpness of the minima in the loss landscape and a method flattening such minima is presented as a remedy. Built on a similar intuition, heuristics such as smoothing the weights or the logit output of neural networks are proposed in Chen et al. (2021). The work of Singla et al. (2021) suggests that the robust overfitting is related to the curvature of the activation functions and that low curvature in the activation function appears to improve robust generalization. In (Dong et al., 2021a), the authors observe the existence of label noise in PGD-AT and regard it as a source of robust overfitting phenomenon, where the label noise refers to that after adversarial perturbation, the original label may no longer reflect the semantics of the example perfectly. The work of Dong et al. (2021b) attributes robust overfitting to a memorization effect and label noise in PGD-AT, and subsequently proposes a mitigation algorithm based on an analysis of memorization. The authors of Yu et al. (2022) observe that in PGD-AT, fitting the training examples with smaller adversarial loss tend to cause robust overfitting and propose a heuristic to remove a fraction of the low-loss example during training. In Kanai et al. (2023), robust overfitting is attributed to the non-smoothness loss used in AT, and the authors propose a smoothing technique as a solution.

Encouraging as these progresses are, the current understanding of robust overfitting is still arguably far from being conclusive. For example, as pointed out in Hameed & Buesser (2022), the explanations in Dong et al. (2021b) and Yu et al. (2022) appear to conflict to each other: the former attributes the robust overfitting to the model fitting the data with large adversarial loss while the latter claims that fitting the the data with small adversarial loss is the source of robust overfitting. Furthermore, the proposed mitigation techniques so far, although have been shown to improve generalization, only reduce the testing robust error by a few percent. This may imply that robust overfitting can be due to a multitude of sources, the full picture remaining obscure.

Our work aims at further understanding robust overfitting, attempting to obtain insights by inspecting the dynamics of PGD-AT, the most popular iterative training algorithm. The inspiration of our experimental design stems from the recognition that along the iterations in PGD-AT, adversarial perturbation effectively induces a new data distribution, say $\tilde{\mathcal{D}}_t$, at each training step $t$. This distribution, different from the original data distribution $\mathcal{D}$, continuously evolves in a fashion that depends on the current model parameter $\theta_t$, which in turn affecting the updating of $\theta_t$. It is then curious whether certain properties of $\tilde{\mathcal{D}}_t$ or the causes of such properties may be related to robust overfitting. We then conducted a set of experiments, which we call "induced distribution experiments" or IDEs, in which we inspect how well a model trained on samples drawn from $\tilde{\mathcal{D}}_t$ (under standard training) generalizes. Specifically, a set of time steps, or "checkpoints", are selected; at each checkpoint $t$, a model is trained from scratch on samples drawn from $\tilde{\mathcal{D}}_t$ under the standard loss. Figure 1 shows one such result on CIFAR-10 (for more results, see Section 4). In the figure, the yellow curve, indicating the generalization gap for robust error, gradually elevates as PGD-AT proceeds; the red curve, indicating the IDE testing error, follows a similar trend. The consistent trends of the two curves suggest a correlation between the generalization behaviour of the learned model along the PGD-AT trajectory and the IDE test errors. Noting that the IDE testing error at checkpoint $t$ indicates the generalization difficulty of the induced distribution $\tilde{\mathcal{D}}_t$. Thus this observation hints that the perturbation at checkpoint $t$, turning the original distribution $\mathcal{D}$ to $\tilde{\mathcal{D}}_t$, may be fundamentally related to this difficulty and to robust overfitting.

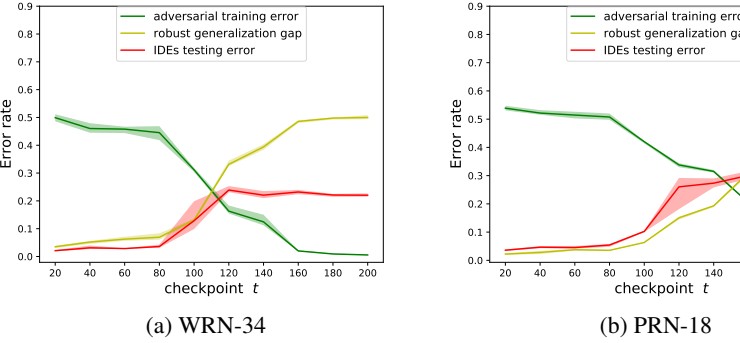

(a) WRN-34                    (b) PRN-18

Figure 1: PGD-AT and the corresponding IDE results of the the Wide ResNet34 (WRN-34) model (Zagoruyko & Komodakis, 2016) and pre-activation ResNet18 (PRN-18) model (He et al., 2016) on the CIFAR-10 dataset. A significant increase in the IDE testing error is observed on both figures with the appearance of robust overfitting, suggesting a correlation between the generalization difficulty on $\tilde{\mathcal{D}}_t$ and robust overfitting.

To further understand the generalization difficulty of $\tilde{\mathcal{D}}_t$ and the impact of perturbation on such difficulty, we derive an upper bound of generalization error for distribution $\tilde{\mathcal{D}}_t$. The bound reveals that a key quantity governing the generalization difficulty is a local "dispersion property" of the adversarial perturbation at checkpoint $t$: less dispersive perturbations provide better generalization guarantees. This result is corroborated by further empirical observations: when robust overfitting occurs, the adversarial perturbations become increasingly dispersive along the PGD-AT trajectory.

We also conducted additional experiments to examine the local dispersion of the perturbations along PGD-AT steps. Interestingly, as PGD-AT proceeds, although the perturbation decreases it magnitudes, its directions appear to further spread out. This observation, clearly correlating with the local

dispersivenss of adversarial perturbation, is, to our best knowledge, reported for the first time. It may open new directions in furthering the understanding of robust overfitting.

This paper opens a new dimension in the study of robust overfitting. Not only have we discovered the important roles of the evolving dispersiveness of adversarial perturbation in robust overfitting, this work has also made us believe that, for indepth understanding of robust overfitting, the dynamics of adversarial training can not be factored out.

## 2 OTHER RELATED WORKS

Our work lies in the topic of robust generalization. Different from the standard setting, the robust generalization in deep learning, especially on high dimensional data, seems significantly difficult. Various work have attempted to understand the reason behind. The work in Schmidt et al. (2018) proves that on simple data models such as the Gaussian and Bernoulli model, the robust generalization can be much harder compared to the standard generalization in the sense of sample complexity. The sample complexity of robust generalization is further explored by using conventional statistical learning tools, such as Rademacher complexity (Khim & Loh (2019), Yin et al. (2018), Awasthi et al. (2020), Xiao et al. (2022a), Attias et al. (2018)), VC dimension (Montasser et al. (2019)) and algorithmic stability analysis (Xing et al. (2021), Xiao et al. (2022b)), and by investigating the problem under the PAC learning frameworks (Cullina et al. (2018), Diochnos et al. (2019)).

The robust generalization has also been theoretically explored beyond the perspective of sample complexity. The work of Li et al. (2022) explain robust generalization from the viewpoint of neural network expressive power. They show that the expressive power of practical models may be inadequate for achieving low robust test error. Li et al. (2019) attempts to understand robust generalization by exploring the inductive bias in gradient descend under the adversarial training setup. Another line of work attempt to understand the overfitting in AT by linking AT with distributional robust optimization (DRO)(Kuhn et al., 2019; Sinha et al., 2020) . The work of Staib & Jegelka (2017) and Bui et al. (2022) demonstrate that different AT schemes can be reformulated as special cases in DRO. The work of Bennouna et al. (2023) shows that under a saddle-point assumption, AT will always cause an at least larger overfitting gap than directly solving an ERM using the standard loss on the data that are adversarially perturbed w.r.t the model obtained by AT. Numerous endeavors have been undertaken to address the challenge of robust overfitting with various empirical training algorithms proposed. The paper Bai et al. (2021) and Qian et al. (2022) provide a comprehensive overview of the latest developments in empirical research in this field.

## 3 ADVERSARIAL TRAINING AND INDUCED DISTRIBUTIONS

We consider a classification setting with input space $\mathcal{X} \subseteq \mathbb{R}^d$ and label space $\mathcal{Y} := \{1, 2, \cdots, K\}$. We will use $\mathcal{D}$ to denote the data distribution on $\mathcal{X} \times \mathcal{Y}$. Let $\Theta \subseteq \mathbb{R}^n$ be the model parameter space. For every $\theta \in \Theta$, let $l_\theta : \mathcal{X} \times \mathcal{Y} \to \mathbb{R}_+$ denote a loss function, where $l_\theta(x, y)$ is the loss (e.g, the cross-entropy loss) of $(x, y)$ for the model with parameter $\theta$. The adversarial population error $R_{\mathcal{D}}^{\mathrm{adv}}(\theta)$ is then

$$R_{\mathcal{D}}^{\mathrm{adv}}(\theta) := \mathbb{E}_{(x,y) \sim \mathcal{D}} \left[ \max_{v \in \mathbb{B}(x,\epsilon)} l_\theta(v, y) \right] \tag{1}$$

where we have chosen $\mathbb{B}(x, \epsilon) := \{t \in \mathbb{R}^d : \|t - x\|_\infty \leq \epsilon\}$ as the $\infty$-norm ball around $x$ with the radius $\epsilon$. In adversarial training, the ultimate objective is to find a model parameter $\theta$ that minimizes $R_{\mathcal{D}}^{\mathrm{adv}}(\theta)$. Having no access to $\mathcal{D}$, in practice, a natural choice is to minimize an empirical version of $R_{\mathcal{D}}^{\mathrm{adv}}(\theta)$, namely, on a training set $S := \{(x_i, y_i)\}_{i=1}^m$ drawn i.i.d from $\mathcal{D}$, solve

$$\min_{\theta \in \Theta} R_S^{\mathrm{adv}}(\theta), \quad \text{where} \quad R_S^{\mathrm{adv}}(\theta) := \frac{1}{m} \sum_{i=1}^m \max_{v_i \in \mathbb{B}(x_i,\epsilon)} l_\theta(v_i, y_i) \tag{2}$$

The most popular adversarial training technique for neural networks for solving this problem is iterating between solving the inner maximization via $k$-step projected gradient descend (PGD) and updating $\theta$ through stochastic gradient descent. We now give a concise explanation of this procedure, referred to as PGD-AT, restating the procedure in Madry et al. (2019).

$k$**-step PGD** A $k$-step PGD can be described by $k$-fold composition of an one-step PGD mapping. With a fixed choice of $x \in \mathcal{X}$, $y \in \mathcal{Y}$, $\theta \in \Theta$, the one-step PGD mapping $\mathcal{A}_{x,y,\theta} : \mathbb{R}^d \to \mathbb{B}(x,\epsilon)$ is defined as

$$\mathcal{A}_{x,y,\theta}(x') := \Pi_{\mathbb{B}(x,\epsilon)} \left[ x' + \lambda \text{sgn} \left( \nabla_{x'} l_\theta(x', y) \right) \right] \tag{3}$$

Here $\Pi_{\mathbb{B}(x,\epsilon)} : \mathbb{R}^d \to \mathbb{B}(x,\epsilon)$ denotes the operation of projecting onto the set $\mathbb{B}(x,\epsilon)$ and $\lambda \in \mathbb{R}_+$ is a hyperparameter. The sgn$(\cdot)$ function takes the sign of a vector on an element-wise basis. The $k$-step PGD mapping $\mathcal{Q}_{x,y,\theta} : \mathbb{R}^d \to \mathbb{B}(x,\epsilon)$ is then

$$\mathcal{Q}_{x,y,\theta}(x') := \left( \underbrace{\mathcal{A}_{x,y,\theta} \circ \cdots \circ \mathcal{A}_{x,y,\theta}}_{k \text{ times}} \right)(x') \tag{4}$$

For later use, we denote the perturbation family $\{\mathcal{Q}_{x,y,\theta} : (x,y) \in \mathcal{X} \times \mathcal{Y}\}$ by $\mathcal{Q}_\theta$.

**Iterations of PGD-AT** In PGD-AT, the process of generating a perturbed example $(v,y)$ from an example $(x,y)$ w.r.t a model parameter $\theta$ can be described as

$$v = \mathcal{Q}_{x,y,\theta}(x + \rho) \tag{5}$$

where $\rho$ is drawn from $\mathcal{U}([-\epsilon, +\epsilon]^d)$, the uniform distribution over the $d$-dimensional cubic $[-\epsilon, +\epsilon]^d$. At iteration $t$, where the model parameter is $\theta_t$, the solution of the inner maximization $\max_{v \in \mathbb{B}(x_i,\epsilon)} l_{\theta_t}(x_i, y_i)$ is taken as $l_{\theta_t}(v_i, y_i)$, and the model parameter is updated by

$$\theta_{t+1} = \theta_t - \eta \nabla_{\theta_t} \left[ \frac{1}{m} \sum_{i=1}^m l_{\theta_t}(v_i, y_i) \right] \tag{6}$$

It is worth noting that the stochastic mapping (5) that perturbs $(x,y)$ to $(v,y)$ depends on $\theta$. Thus the distribution of the random variable pair $(v,y) = (\mathcal{Q}_{x,y,\theta_t}(x + \rho), y)$ at iteration $t$ depends on $\theta_t$. We will denote this distribution by $\tilde{\mathcal{D}}_{\theta_t}$, or simply $\tilde{\mathcal{D}}_t$, and refer to it as the *(adversarial training) induced distribution* at iteration $t$. Then at iteration $t$, we may regard the perturbed examples $\{(v_i, y_i)\}$ as an i.i.d. sample from $\tilde{\mathcal{D}}_t$.

Note that the distribution $\tilde{\mathcal{D}}_t$ evolves with $\theta_t$ during training and in turn affects the update of $\theta_t$. Since perturbation at every iteration is restricted to a small neighborhood of $x$, $\tilde{\mathcal{D}}_t$ is "not far" from the original data distribution $\mathcal{D}$. However, in our following experiments, we show that the induced distributions $\tilde{\mathcal{D}}_t$ can be much harder to generalize than the original distribution $\mathcal{D}$, particularly when robust overfitting occurs.

## 4 TRAINING ON THE INDUCED DISTRIBUTIONS

The following experiments are conducted. First PGD-AT is performed on a training set $S$. Along this process, for a prescribed set of training iterations (or "checkpoints") $t = k_1, k_2, \ldots, k_N$, the model parameter $\theta_t$ at each checkpoint $t$ is saved. Then at each checkpoint $t$, each example in the training set $S$ is perturbed according to (5) with $\theta = \theta_t$, giving rise to the perturbed training set $\tilde{S}_t$. The testing set $T$ is also similarly perturbed, giving rise to perturbed testing set $\tilde{T}_t$. The model is then retrained fully on $\tilde{S}_t$, using standard training (i.e, without perturbation) with random initialization. The resulting model is tested on $\tilde{T}_t$. Note that in this setting, both $\tilde{S}_t$ and $\tilde{T}_t$ are i.i.d. samples from $\tilde{\mathcal{D}}_t$. We call these experiments "induced distribution experiment" (IDE) for the ease of reference.

The experiments are conducted on MNIST (LeCun et al., 1998), CIFAR10 and CI-FAR100(Krizhevsky et al., 2009). We also conduct experiments on a "scaled-down" version of the ImageNet dataset (Russakovsky et al., 2015). Given that PGD-AT is known to be significantly challenging and computationally expensive on the full-scale ImageNet dataset, we draw inspiration from the approach presented in Tsipras et al. (2019) and made a Reduced ImageNet by aggregating several subsets of the original ImageNet. Our reduced ImageNet comprises 10 classes, each containing 5000 training samples and approximately 1000 testing samples per class. More details concerning this dataset are given in Appendix A. We use the following settings for PGD-AT: For

MNIST, following the settings in Madry et al. (2019), we train a small CNN model using 40-step PGD with step size $\lambda = 0.01$ and perturbation radius $\epsilon = 0.3$. For the other three datasets, we train the pre-activation ResNet (PRN) model (He et al., 2016) and the Wide ResNet (WRN) model (Zagoruyko & Komodakis, 2016). We use 5-step PGD with $\epsilon = 4/255$ for the Reduced ImageNet and 10-step PGD with $\epsilon = 8/255$ for CIFAR-10 and CIFAR-100 according to Rice et al. (2020) in PGD-AT. We set $\lambda = 2/255$ on CIFAR10 and CIFAR100, $\lambda = 0.9/255$ on the reduced ImageNet. More details concerning the hyperparameter settings are given in Appendix A.

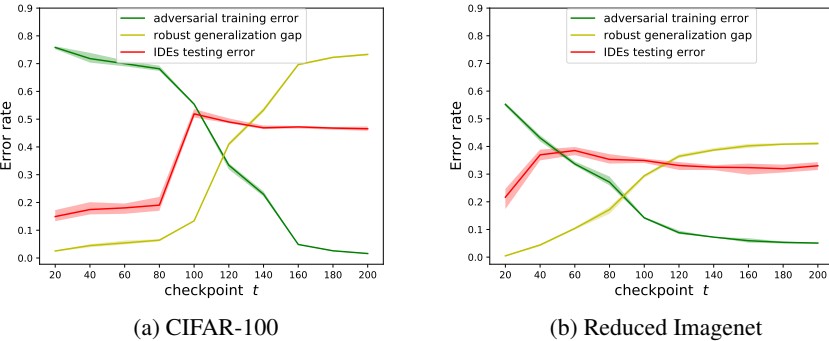

| (a) CIFAR-100 | (b) Reduced Imagenet |

Figure 2: PGD-AT and the corresponding IDE results on CIFAR-100 and the reduced ImageNet. The behaviour of the red curves matches that of the yellow curve, as we observe a substantial rise in IDE testing errors concurrent with the emergence of robust overfitting. This demonstrate a compelling correlation between these two quantities.

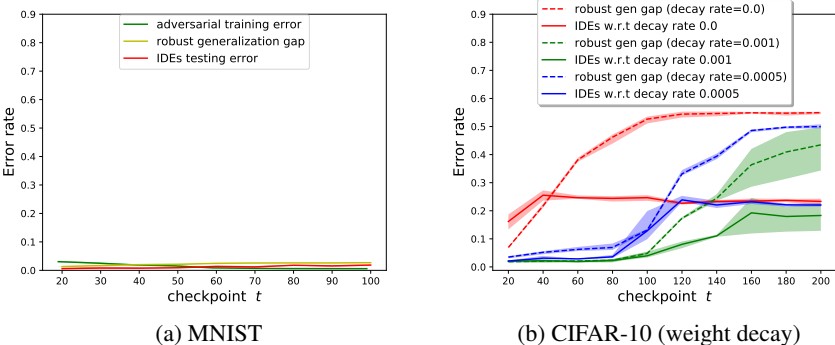

| (a) MNIST | (b) CIFAR-10 (weight decay) |

Figure 3: (a)PGD-AT and the corresponding IDE results on MNIST. The experiment presents a scenario that a good robust generalization is achieved. The absence of robust overfitting coincides with the consistently low IDE testing error. (b) exhibits the outcomes of additional experiments conducted on CIFAR-10. In the experiments, we perform PGD-AT with various weight decay rates and conduct IDEs for each of the PGD-AT variant. The blue curves are reproduced from Figure1 (a), serving as a reference for a clear comparison. The results further solidify the correlation between the robust overfitting and the IDE testing error.

For each dataset, the experiments are repeated five times with different random seeds. The experimental results on CIFAR-10 and CIFAR-100 are shown in Figure 1 and 2 (a), where the green and yellow curves illustrate a phenomenon known as robust overfitting (Rice et al., 2020): after a certain point in PGD-AT, the robust generalization gap (yellow curves), that is $|R_{\mathcal{D}}^{\mathrm{rob}}(\theta_t) - R_S^{\mathrm{rob}}(\theta_t)|$, steadily increases while the adversarial training error (green curves) constantly decreases. The red curves in the figures depict the testing errors of IDEs w.r.t various checkpoints in PGD-AT. Notably, a significant rise in the average IDE testing error is observed between the 80th and 120th checkpoints, increasing from 3.6% to 23.89% for CIFAR-10 with WRN and from 19% to 48.99% for CIFAR-100. Furthermore, this shift coincides with the onset of robust overfitting, where a significant rise in robust generalization gap is also observed. These results indicate that as $\theta_t$ evolves along

PGD-AT, the model trained on $\tilde{\mathcal{D}}_t$ becomes harder to generalize. More importantly, this difficulty in generalization appears to be closely linked to the robust overfitting phenomenon. This potential connection is further demonstrated by the experimental results on the reduced ImageNet (see Figure2 (b)), where robust overfitting emerges at an earlier training stage and simultaneously a rise in the IDE testing error occurs. This increment in the IDE testing error is also substantial, with an averaged error of 21.65% at the 20th checkpoint elevating to 38.52% at the 60th checkpoint.

Our experiments on MNIST (see Figure 3 (a)) exhibits a scenario where a good robust generalization is achieved. Interestingly, a small IDE testing error is maintained throughout the evolution of $\tilde{\mathcal{D}}_t$ with the absence of robust overfitting. Figure 3 (b) shows results from additional experiments on CIFAR-10. In these experiments, we perform PGD-AT with different level of weight decay to control the level of robust overfitting. Subsequently, IDEs are conducted for each such variant of PGD-AT. In Figure 3 (b), each distinct color corresponds to a different weight decay factor utilized in PGD-AT. Within each color category, the dashed curves and the corresponding solid lines represent, respectively, the robust generalization gaps and the IDE results associated with that specific PGD-AT variant. As anticipated, increasing the weight decay factor results in a notable reduction in the robust generalization gap, while conversely, decreasing the weight decay factor leads to the opposite effect. This is shown by the downward shift in the dashed curves across the three color categories. Additionally, a clear synchronization can be observed between each pair of dashed and solid curves, with lower dashed curves consistently corresponding to lower solid curves in the same color category.

At this end, we have established that the increasing difficulty of generalization inherent in the induced distribution plays an important role in robust overfitting. These observations highlight the impact of the dynamics of PGD-AT on robust overfitting, beyond the static quantities, such as loss landscape.

## 5 GENERALIZATION PROPERTIES OF THE INDUCED DISTRIBUTIONS

The experiments above reveals an interesting phenomenon that as $\tilde{\mathcal{D}}_t$ evolves in PGD-AT, the model obtained from $\tilde{\mathcal{D}}_t$ becomes harder to generalize, particularly when robust overfitting occurs. This suggests that the increasing generalization difficulty of the induced distribution $\tilde{\mathcal{D}}_t$ along PGD-AT trajectory contributes as an important factor to robust overfitting. It remains curious what causes $\tilde{\mathcal{D}}_t$ to become harder to generalize in adversarial training. Here we provide a theoretical explanation and corroborate it with empirical observations.

There are two views of adversarial examples, "in-distribution view" (e.g., in Song et al. (2017); Gilmer et al. (2018); Raghunathan et al. (2019)) and "out-of-distribution view" (e.g., in Szegedy et al. (2014); Khoury & Hadfield-Menell (2018); Stutz et al. (2019)). In the out-of-distribution view, adversarial examples are considered as living outside the data manifold. In the in-distribution view, adversarial examples are considered as located within the support of the true data distribution but having low probability (density). This paper takes the in-distribution view and make the following assumption on the data distribution $\mathcal{D}$, which we may re-express as the pair $(\mathcal{D}_{\mathcal{X}}, h^*)$, where $\mathcal{D}_{\mathcal{X}}$ is the marginal distribution of $\mathcal{D}$ on the input space $\mathcal{X}$ and $h^*$ is the ground-truth classifier: There is a distribution $\mathcal{D}_{\mathcal{X}}^*$ on $\mathcal{X}$ and some small $\epsilon > 0$, such that for every $x$ in the support of $\mathcal{D}_{\mathcal{X}}^*$, $h^*(x) = h^*(x+\rho)$ for all $\|\rho\|_\infty \leq \epsilon$, and $D_{\mathcal{X}}$ is the convolution of $\mathcal{D}_{\mathcal{X}}^*$ with the uniform distribution on $\mathbb{B}(0, \epsilon)$. For the simplicity of notation, we write the $(x, h^*(x))$ as $(x, y)$, and when $x$ is drawn from $\mathcal{D}_{\mathcal{X}}^*$, we denote the joint distribution of $(x, y)$ by $\mathcal{D}^*$.

We now consider a standard classification problem with $\tilde{\mathcal{D}}_\theta$ as the underlying data distribution over $\mathcal{X} \times \mathcal{Y}$. Let $\mathcal{H}$ be a hypothesis class for this learning problem, where each member $h \in \mathcal{H}$ is a function mapping $\mathcal{X}$ to $\mathcal{Y}$. Note that the hypothesis class $\mathcal{H}$ may have not be related to the model used for adversarial training in any way. Let $\ell : \mathcal{Y} \times \mathcal{Y} \to \mathbb{R}$ be a non-negative loss function. For every $h \in \mathcal{H}$, let function $f_h : \mathcal{X} \times \mathcal{Y} \to \mathbb{R}$ be defined by $f_h(x, y) := \ell(h(x), y)$. Let $\mathcal{F} := \{f_h : h \in \mathcal{H}\}$. From here on, we will restrict our attention to this "loss hypothesis class" $\mathcal{F}$ and study how well the members of $\mathcal{F}$ generalize from an i.i.d. training sample $\{(v_i, y_i)\}_{i=1}^m$ drawn from $\tilde{\mathcal{D}}_\theta$ to the unknown distribution $\tilde{\mathcal{D}}_\theta$. Specifically, for each $f \in \mathcal{F}$, the key quantity of our interest is the *generalization gap* $\left| \frac{1}{m} \sum_{i=1}^m f(v_i, y_i) - \mathbb{E}_{(v,y) \sim \tilde{D}_\theta} f(v, y) \right|$.

As we will soon show, the generalization gap turns out to be related to a key quantity characterizing a local property of the perturbation map $Q_{x,y,\theta}$, through which $\tilde{D}_\theta$ is induced.

**Definition 1.** Let $(\mathcal{X}', \|\cdot\|_2)$ be a norm space equipped with the 2-norm. Given a map $T : \mathcal{X} \to \mathcal{X}'$ and an arbitrary bounded measurable subset $C$ of $\mathcal{X}$, we define the $C$-dispersion of $T$ by

$$\gamma_C(T) := \mathbb{E}_{x,x' \sim \mathcal{U}(C)} \|T(x) - T(x')\|_2^2 \tag{7}$$

where $\mathcal{U}(C)$ denotes a uniform distribution over $C$. Intuitively, this quantity measures on average how far two random points in $C$ spread after being mapped by $T$. Now restricting $T = Q_{x,y,\theta}$ and $C = \mathbb{B}(x, \epsilon)$, we have

$$\gamma_{\mathbb{B}(x,\epsilon)}(Q_{x,y,\theta}) = \mathbb{E}_{\rho,\rho' \sim \mathcal{U}([-\epsilon,+\epsilon]^d)} \|Q_{x,y,\theta}(x+\rho) - Q_{x,y,\theta}(x+\rho')\|_2^2.$$

For simplicity we rewrite this quantity as $\tilde{\gamma}_\theta(x, y)$ and refer to it as the *local dispersion* of the perturbation (family) $Q_\theta$ at $(x, y)$.

Notably, for any given $(x, y)$, it is possible to estimate this quantity by Monte-Carlo approximation. In our experiments, to estimate local dispersion $\tilde{\gamma}_\theta(x, y)$, we sample 250 pairs of $(\rho, \rho')$ and approximate the expectation by the sample mean.

We now show that the expected value of local dispersion of $Q_\theta$ is a governing term of the generalization gap, under the following conditions.

- (Lipchitzness of $f$ over $\mathcal{X}$) For any $y \in \mathcal{Y}$, $|f(x, y) - f(x', y)| \le \beta\|x - x'\|_2$ for $\forall x, x' \in \mathcal{X}$.

- (Loss boundedness) $\sup\limits_{x,y \in \mathcal{X} \times \mathcal{Y}} |f(x, y)| = B < \infty$.

- (Boundedness of perturbation-smoothed loss) $\sup\limits_{x,y \in \mathrm{supp}(\mathcal{D})} |\mathbb{E}_\rho f(Q_{x,y,\theta}(x+\rho), y)| = A < \infty$, where $\mathrm{supp}(\mathcal{D})$ denote the support of distribuion $\mathcal{D}$.

**Theorem 1.** Let $f \in \mathcal{F}$ and suppose that $f$ satisfies the above conditions. Then for any $\tau > 0$, with probability at least $1 - \tau$ over the i.i.d. draws of sample $\{(v_i, y_i)\}_{i=1}^m$ from $\tilde{D}_\theta$,

$$\left| \frac{1}{m} \sum_{i=1}^m f(v_i, y_i) - \mathbb{E}f(v, y) \right| \le \frac{2\beta}{\sqrt{m}}\sqrt{\mathbb{E}_{\mathcal{D}^*}\tilde{\gamma}_\theta(x, y)} + \frac{2A}{\sqrt{m}} + 2B\sqrt{\frac{\log\frac{1}{\tau}}{2m}} \tag{8}$$

We leave the proof in Appendix B. The theorem shows that the generalization gap of any $f$ w.r.t to the distribution $\tilde{D}_\theta$ is affected by the expected local dispersion (ELD) $\mathbb{E}_{\mathcal{D}^*}\tilde{\gamma}_\theta(x, y)$ of $Q_\theta$. More specifically, it suggests that a small generalization gap can be achieved when $\mathbb{E}_{\mathcal{D}^*}\tilde{\gamma}_\theta(x, y)$ is small.

To validate the usefulness of this theorem, we performed experiments to estimate the ELD $\mathbb{E}_{\mathcal{D}^*}\tilde{\gamma}_\theta(x, y)$ for $\theta$ at various checkpoints $t$. Note that the expectation here is over the distribution $\mathcal{D}^*$, from which no samples are available. However, due to the relationship between $\mathcal{D}_\mathcal{X}$ and $\mathcal{D}_\mathcal{X}^*$, namely that $\mathcal{D}_\mathcal{X}$ is merely an $\epsilon$-smoothed version of $\mathcal{D}_\mathcal{X}^*$, one expects that when we draw $x$ from $\mathcal{D}$, $\mathcal{D}_\mathcal{X}(x) \approx \mathcal{D}_\mathcal{X}^*(x)$ with high probability. Then when we estimate $\mathbb{E}_{\mathcal{D}^*}\tilde{\gamma}_\theta(x, y)$ using an i.i.d. sample $S$ drawn from $\mathcal{D}$, we may approximately regard $S$ as drawn i.i.d. from $\mathcal{D}^*$, and estimate $\mathbb{E}_{\mathcal{D}^*}\tilde{\gamma}_\theta(x, y)$ by $\mathbb{E}_{\mathcal{D}^*}\tilde{\gamma}_\theta(x, y) \approx \frac{1}{m}\sum_{i=1}^m \tilde{\gamma}_\theta(x_i, y_i)$, where $\{(x_i, y_i)\}$ are drawn from $\mathcal{D}$. Notably this is a slightly biased estimator of $\mathbb{E}_{\mathcal{D}^*}\tilde{\gamma}_\theta(x, y)$, and the bias decreases with $\epsilon$.

In our experiments, we inspect the evolution of the distribution of $\tilde{\gamma}_{\theta_t}(x, y)$ (or $\tilde{\gamma}_t(x, y)$ for simplicity) along the PGD-AT trajectory. Figure 4(a) plots the histogram of $\tilde{\gamma}_t(x, y)$ for the testing set of CIFAR-10 at three different PGD-AT checkpoints. It is clear that the distribution shifts to the right as PGD-AT proceeds, indicating that the perturbation operator $Q_{\theta_t}$ becomes more locally dispersive as training goes on. We then estimate ELD on the testing set (the green curve in Figure 4(b)). We plot one of the IDE results using red curve in the same figure for a clearer comparison. As shown in the figure, ELD is getting larger along PGD-AT, correlating with the increasing difficulty of generalization on $\tilde{D}_t$. These results suggests that the bound in Theorem 1 adequately characterizes the generalization behaviour in the IDE experiments, which in turn suggests that the local dispersiveness of the perturbation operator contributes to robust overfitting in PGD-AT. Similar experimental results are also observed across different datasets (see Appendix C).

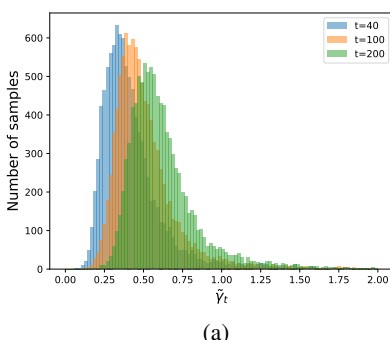
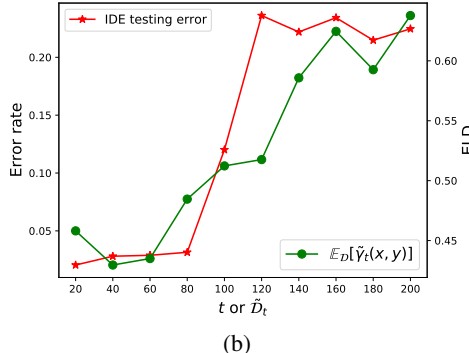

(a)                                                (b)

Figure 4: Experiments on the CIFAR-10 testing set: (a) histograms of $\tilde{\gamma}_t(x,y)$ at three distinct PGD-AT checkpoints. (b) The evolution of ELD w.r.t $t$ and the IDE testing error for each $\tilde{\mathcal{D}}_t$. The results show that the level of $\tilde{\gamma}_t(x,y)$ increases during PGD-AT and correspondingly the model obtained from $\tilde{\mathcal{D}}_t$ becomes harder to generalize. This implies that the local property of $\mathcal{Q}_{x,y,\theta_t}$ characterized by $\tilde{\gamma}_t(x,y)$ plays a dominate role in influencing the generalization of $\tilde{\mathcal{D}}_t$.

## 6 OTHER IMPLICATIONS

Our preceding theoretical analysis underscores the critical role played by the local properties of $\mathcal{Q}_{x,y,\theta_t}$ in affecting the generalization performance for $\tilde{\mathcal{D}}_t$. This, in turn, inspires our curiosity to investigate whether additional local properties, beyond local dispersion, also posses critical influences on the generalization of $\tilde{\mathcal{D}}_t$. As such, we inspect the expected distance between the adversarial examples generated by PGD and its clean counterparts, defined as

$$d_\theta(x,y) := \mathbb{E}_{\rho \sim \mathcal{U}([-\epsilon,\epsilon]^d)} \|\mathcal{Q}_{x,y,\theta}(x+\rho) - x\|_2 \tag{9}$$

By triangle inequality, we notice that

$$\mathbb{E}_{\rho,\rho' \sim \mathcal{U}([-\epsilon,+\epsilon]^d)} \|\mathcal{Q}_{x,y,\theta}(x+\rho) - \mathcal{Q}_{x,y,\theta}(x+\rho')\|_2 \tag{10}$$

$$=\mathbb{E}_{\rho,\rho' \sim \mathcal{U}([-\epsilon,+\epsilon]^d)} \|\mathcal{Q}_{x,y,\theta}(x+\rho) - x + x - \mathcal{Q}_{x,y,\theta}(x+\rho')\|_2 \tag{11}$$

$$\leq 2d_\theta(x,y) \tag{12}$$

The term (10) is related to the local dispersion of $\mathcal{Q}_{x,y,\theta}$ despite that the definition of the local dispersion computes the square of the $l_2$-distance. Recall that we have observed an increase in the level of local dispersion along the PGD-AT trajectory. According to the inequality (12), one might logically expect that the level of $d_\theta(x,y)$ should also increase, meaning that the perturbed data generated by $x$ are getting not only more "dispersed" around $x$ but also move farther from $x$. However, our experimental findings present a contradictory result. Instead of an increase, we observe a decrease in the level of $d_\theta(x,y)$ during PGD-AT. This unexpected trend suggests that the perturbed data generated by $x$ are, in fact, moving closer to the original data point $x$.

In our experiments, we estimate $d_\theta(x,y)$ by computing the sample mean with 250 samples of $\rho$ drawn from $\mathcal{U}([-\epsilon,\epsilon]^d)$. We analyze the dynamic behavior of $d_{\theta_t}(x,y)$, which we refer to as $d_t(x,y)$ for simplicity, along the PGD-AT trajectory. In Figure 5 (a), we present histogram of $d_t(x,y)$ for the CIFAR-10 testing set at three distinct training checkpoints. Notably, the histogram exhibits a notable mode shift towards a smaller value, indicating a trend that as PGD-AT proceeds, the generated adversarial examples progressively approach their clean counterparts. The reduction in the level of $d_t(x,y)$ along PGD-AT is further observed by evaluating the expectation $\mathbb{E}_{\mathcal{D}} d_t(x,y)$ on the testing set (see Figure 5 (c), green curve), where a clear drop in $\mathbb{E}_{\mathcal{D}} d_t(x,y)$ is exhibited.

As a reminder, we previously noted that adversarial examples tend to become more dispersed around their clean counterparts as training progresses. The experimental findings presented here shed light on this phenomenon, suggesting that the growing dispersion is likely to be a result of the perturbation angles expanding, while the perturbation magnitudes seem to have a lesser impact on the level of dispersion.

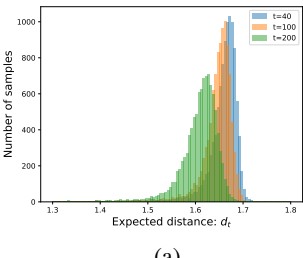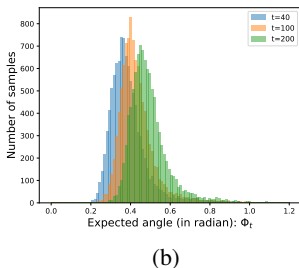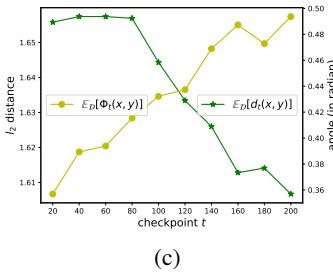

| (a) | (b) | (c) |

Figure 5: Experiments on the CIFAR-10 testing set. (a) and (b): histograms of $d_t(x, y)$ and $\Phi_t(x, y)$ at different adversarial training epochs $t$. (c): The evolution of $\mathbb{E}_{\mathcal{D}} d_t(x, y)$ and $\mathbb{E}_{\mathcal{D}} \Phi_t(x, y)$ along adversarial training trajectory. Combined with the results in Figure 4, an interesting phenomenon in adversarial training is revealed: as the adversarial proceeds, the perturbed data generated by $x$ are getting closer to $x$ and in the meanwhile getting more dispersed potentially due to the spreading of perturbation angles.

To verify this conjecture, we evaluate the expected angle between a pair of perturbations generated from $(x, y)$, defined as

$$\Phi_\theta(x, y) := \mathbb{E}_{\rho, \rho' \sim \mathcal{U}([-\epsilon, +\epsilon]^d)} \cos^{-1} \left( \frac{(\mathcal{Q}_{x,y,\theta}(x + \rho) - x)^T (\mathcal{Q}_{x,y,\theta}(x + \rho') - x)}{\|\mathcal{Q}_{x,y,\theta}(x + \rho) - x\|_2 \|\mathcal{Q}_{x,y,\theta}(x + \rho') - x\|_2} \right) \quad (13)$$

with the expectation estimated by computing the sample mean of 250 pairs of $\rho, \rho'$ drawn from $\mathcal{U}([-\epsilon, +\epsilon]^d)$. Figure 5 (b) plots the histograms of $\Phi_{\theta_t}(x, y)$ (or $\Phi_t(x, y)$) at three distinct checkpoint $t$ and Figure 5 (c) illustrate the evolution of $\mathbb{E}_{\mathcal{D}} \Phi_t(x, y)$ with the yellow curve. The results present an increase in the level of $\Phi_t(x, y)$, indicating a spreading of perturbation angles and less "aligned" perturbations generated by each $x$ during PGD-AT. Similar experimental results have been observed across other datasets. (see Appendix D)

We conjecture that this wider spread of angles in adversarial perturbations is a consequence of an intricate or "ragged" shape in the model's decision boundary. In essence, the shape of the decision boundary has a substantial influence on the direction of perturbations. For instance, the perturbations generated by linear classifiers are always aligned due to that the decision boundary is "smooth". Conversely, one would expect that a jagged or irregular decision boundary could result in perturbations that are both more dispersed and less aligned. We speculate that the presented dynamics of $\mathbb{E}_{\mathcal{D}} \Phi_t(x, y)$ and $\mathbb{E}_{\mathcal{D}} \tilde{\gamma}_t(x, y)$ can be explained as: during the early stages of PGD-AT, the adversarial perturbations generated by the data $(x, y)$ exhibit a higher degree of alignment due to the initial smoothness of the model's decision boundary. This results in a smaller level of $\mathbb{E}_{\mathcal{D}} \Phi_t(x, y)$ and $\mathbb{E}_{\mathcal{D}} \tilde{\gamma}_t(x, y)$. However, as training progresses, the decision boundary is twisted, in order to fit or "memorize" the training data, causing the perturbations to become less aligned and more dispersed, leading to the rise in the level of $\mathbb{E}_{\mathcal{D}} \Phi_t(x, y)$ and $\mathbb{E}_{\mathcal{D}} \tilde{\gamma}_t(x, y)$. Consequently, the increasing dispersion or spread of angles may serve as an indicative measure for the degree of irregularity present in the decision boundary.

Our observation of the increasing dispersion and spreading angles of adversarial perturbations along PGD-AT is, to our best knowledge, a novel finding. This discovery may provide valuable insights into comprehending the dynamic of PGD-AT and the phenomenon of robust overfitting.

## 7 CONCLUSION

In this paper, we show that adversarial perturbation induced distribution plays an important role in robust overfitting. In particular, we observe experimentally that the increasing generalization difficulty of the induced distribution along the training trajectory is correlated with robust overfitting. Our theoretical analysis suggests that a key factor governing this difficulty is the local dispersion of the perturbation. Experimental results confirm that as adversarial training proceeds, the perturbation becomes more dispersed, validating our theoretical results. Various additional insights are also presented.

This work points to a new direction in the search of explanations for robust overfitting. Remarkably, through this work, we demonstrate that the trajectory of adversarial training plays an important role in robust overfitting. Studying the dynamics of adversarial training is arguably a promising approach to developing deeper understanding of this topic. In particular, we speculate that studying the effect of gradient-based parameter update may provide additional insight.

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

## A    DETAILED EXPERIMENTAL SETUP

Our Reduced ImageNet is made by aggregating several semantically similar subsets of the original ImageNet, resulting in a total of 66594 images. This dataset is then partitioned into a training set containing 5,000 images per class and a testing set containing approximately 1,000 images per class. Compared to the restricted ImageNet in Russakovsky et al. (2015), our dataset has a more balanced sample size across each classes. Table 1 illustrates the specific classes from the original ImageNet that have been aggregated in our dataset.

For adversarial training, the settings on different datasets are summarized in Table 2. Data augmentation is performed on these datasets except for MNIST. during the training. For CIFAR-10 and CIFAR-100 we follow the data augmentation setting in Rice et al. (2020). For our reduced ImageNet, we adopt the same data augmentation scheme that is used on the restricted ImageNet in Yang et al. (2020).

| Classes in the reduced ImageNet | Classes in ImageNet |
|---|---|
| "dog" | 86 to 90 |
| "cat" | (8,10,55,95,174) |
| "truck" | 279 to 283 |
| "car" | 272 to 276 |
| "beetles" | 623 to 627 |
| "turtle" | 458 to 462 |
| "crab" | 612 to 616 |
| "fish" | 450 to 454 |
| "snake" | 477 to 481 |
| "spider" | 604 to 608 |

Table 1: The left column presents the classes within our reduced ImageNet dataset, with each class being an aggregation of the corresponding classes from the full-scale ImageNet dataset, as depicted in the right column.

| | MNIST | CIFAR-10 | CIFAR-100 | Reduced ImageNet |
|---|---|---|---|---|
| model | small CNN | PRN18&WRN-34 | WRN-34 | PRN-50 |
| optimizer | Adam | SGD | SGD | SGD |
| weight deacy | None | $5 \times 10^{-4}$ | $5 \times 10^{-4}$ | None |
| batch size | 128 | 128 | 128 | 128 |
| $\epsilon$ | 0.3 | 8/255 | 8/255 | 4/255 |
| $\lambda$ | 0.01 | 2/255 | 2/255 | 0.9/255 |
| number of PGD | 40 | 10 | 10 | 5 |

Table 2: Settings in adversarial training across different datasets

For the IDEs on each datasets, the settings are outlined in Table 3. It is important to note that for each of the individual IDEs conducted on the same dataset, we maintain consistent training settings. This includes using the same model architecture with identical model size and the same level of regularization. This ensures a fair comparison of the IDE results obtained from the same dataset. Furthermore, the model is trained to achieve zero training error in all the IDEs, excluding the situation that the degeneration in model performance could be attributed to inadequate training procedures.

| | MNIST | CIFAR-10 | CIFAR-100 | Reduced ImageNet |
|---|---|---|---|---|
| model | small CNN | WRN-34 | WRN-34 | PRN-50 |
| optimizer | Adam | SGD | SGD | SGD |
| weight deacy | None | $5 \times 10^{-4}$ | $5 \times 10^{-4}$ | $5 \times 10^{-4}$ |
| batch size | 128 | 128 | 128 | 128 |

Table 3: Settings in the IDE across different datasets

# B PROOF OF THE THEOREM

We use the notations introduced in the main text. For shorter notations, let $z = (x, y)$ with $x$ following the distribution $\mathcal{D}_{\mathcal{X}}^*$ and $y = h^*(x)$. Let $u = (v, y)$ and $f(u) := f(v, y)$. We write $\mathcal{Q}_{x,y,\theta}$ as $\mathcal{Q}_z$, since our derivation does not explicitly depend on the choice of $\theta$.

Denote by $g(u_1 \cdots u_m) := \left| \frac{1}{m} \sum_{i=1}^{m} f(u_i) - \mathbb{E}f(u) \right|$. We have for any $1 \leq j \leq m$

$$\sup_{u_1, \cdots, u_m, u_j'} \left| g(u_1, \cdots, u_m) - g(u_1, \cdots, u_j', u_{j+1}, \cdots u_m) \right| \tag{14}$$

$$= \sup_{u_1, \cdots, u_m, u_j'} \left\| \left| \frac{1}{m} \sum_{i=1}^{m} f(u_i) - \mathbb{E}f(u) \right| - \left| \frac{1}{m} \left( \sum_{i=1, i \neq j}^{m} f(u_i) + f(u_j') \right) - \mathbb{E}_u f(u) \right| \right\| \tag{15}$$

$$\leq \sup_{u_1, \cdots, u_m, u_j'} \left| \frac{1}{m} \sum_{i=1}^{m} f(u_i) - \mathbb{E}_u f(u) - \frac{1}{m} \left( \sum_{i=1, i \neq j}^{m} f(u_i) + f(u_j') \right) + \mathbb{E}_u f(u) \right| \tag{16}$$

$$= \sup_{u_j, u_j'} \frac{1}{m} \left| f(u_j) - f(u_j') \right| \tag{17}$$

$$\leq \frac{1}{m} \sup_{u_j} |f(u_j)| + \frac{1}{m} \sup_{u_j'} |f(u_j')| \tag{18}$$

$$\leq \frac{2B}{m} \tag{19}$$

where the inequality (16) follows from the inverse triangle inequality. The inequality (18) and (19) make use of the triangle inequality and the boundedness condition of $f$.

With the result derived above, by McDiarmid inequality, we have for all $\mu > 0$

$$\Pr \left[ g(u_1 \cdots u_m) - \mathbb{E}_U g(u_1 \cdots u_m) \geq \mu \right] \leq \exp \left( \frac{-m\mu^2}{B} \right)$$

This is equivalent to saying that with probability $1 - \tau$, we have

$$g(u_1 \cdots u_m) \leq \mathbb{E}_U g(u_1 \cdots u_m) + 2B \sqrt{\frac{\log \frac{1}{\tau}}{2m}} \tag{20}$$

Given this, the following parts aim at constructing an upper bound for the term $\mathbb{E}_U g(u_1 \cdots u_m)$.

For shorter notation, let $U := (u_1, \cdots, u_m)$, $Z := (z_1, \cdots, z_m)$, $\Gamma := (\rho_1, \cdots, \rho_m)$, $F(Z, \Gamma) := \frac{1}{m} \sum_{i=1}^{m} f(\mathcal{Q}_{z_i}(x_i + \rho_i), y_i)$. We have

$$\mathbb{E}_U g(u_1 \cdots u_m) \tag{21}$$

$$= \mathbb{E}_U \left| \frac{1}{m} \sum_{i=1}^{m} f(u_i) - \mathbb{E}f(u) \right| \tag{22}$$

$$= \mathbb{E}_U \left| \frac{1}{m} \sum_{i=1}^{m} f(u_i) - \mathbb{E}_{\hat{U}} \left[ \frac{1}{m} \sum_{i=1}^{m} f(\hat{u}_i) \right] \right| \tag{23}$$

$$\leq \mathbb{E}_U \mathbb{E}_{\hat{U}} \left| \frac{1}{m} \sum_{i=1}^{m} f(u_i) - \frac{1}{m} \sum_{i=1}^{m} f(\hat{u}_i) \right| \tag{24}$$

$$= \mathbb{E}_Z \mathbb{E}_\Gamma \mathbb{E}_{\hat{Z}} \mathbb{E}_{\hat{\Gamma}} \left| \frac{1}{m} \sum_{i=1}^{m} f(\mathcal{Q}_{z_i}(x_i + \rho_i), y_i) - \frac{1}{m} \sum_{i=1}^{m} f(\mathcal{Q}_{\hat{z}_i}(\hat{x}_i + \hat{\rho}_i), \hat{y}_i) \right| \tag{25}$$

$$= \mathbb{E}_Z \mathbb{E}_\Gamma \mathbb{E}_{\hat{Z}} \mathbb{E}_{\hat{\Gamma}} \left| F(Z, \Gamma) - \mathbb{E}_{\bar{\Gamma}} F(Z, \bar{\Gamma}) + \mathbb{E}_{\bar{\Gamma}} F(Z, \bar{\Gamma}) - F(\hat{Z}, \hat{\Gamma}) + \mathbb{E}_{\tilde{\Gamma}} F(\hat{Z}, \tilde{\Gamma}) - \mathbb{E}_{\tilde{\Gamma}} F(\hat{Z}, \tilde{\Gamma}) \right| \tag{26}$$

$$\leq \mathbb{E}_Z \mathbb{E}_\Gamma \left| F(Z, \Gamma) - \mathbb{E}_{\bar{\Gamma}} F(Z, \bar{\Gamma}) \right| + \mathbb{E}_{\hat{Z}} \mathbb{E}_{\hat{\Gamma}} \left| F(\hat{Z}, \hat{\Gamma}) - \mathbb{E}_{\tilde{\Gamma}} F(\hat{Z}, \tilde{\Gamma}) \right| + \mathbb{E}_Z \mathbb{E}_{\hat{Z}} \left| \mathbb{E}_{\bar{\Gamma}} F(Z, \bar{\Gamma}) - \mathbb{E}_{\tilde{\Gamma}} F(\hat{Z}, \tilde{\Gamma}) \right| \tag{27}$$

$$= \underbrace{2 \mathbb{E}_Z \mathbb{E}_\Gamma \left| F(Z, \Gamma) - \mathbb{E}_{\bar{\Gamma}} F(Z, \bar{\Gamma}) \right|}_{①} + \underbrace{\mathbb{E}_Z \mathbb{E}_{\hat{Z}} \left| \mathbb{E}_{\bar{\Gamma}} F(Z, \bar{\Gamma}) - \mathbb{E}_{\tilde{\Gamma}} F(\hat{Z}, \tilde{\Gamma}) \right|}_{②} \tag{28}$$

where (24) follows from Jensen's inequality and (27) is by the triangle inequality. We now individually construct upper bounds for the term ① and ②.

For the term ①, we have

$$2\mathbb{E}_Z\mathbb{E}_\Gamma \left| F(Z,\Gamma) - \mathbb{E}_{\bar{\Gamma}}F(Z,\bar{\Gamma})\right| \tag{29}$$

$$\leq 2\mathbb{E}_Z\mathbb{E}_\Gamma\mathbb{E}_{\bar{\Gamma}} \left| F(Z,\Gamma) - F(Z,\bar{\Gamma})\right| \tag{30}$$

$$=2\mathbb{E}_Z\mathbb{E}_\Gamma\mathbb{E}_{\bar{\Gamma}} \left| \frac{1}{m}\sum_{i=1}^m f(\mathcal{Q}_{z_i}(x_i+\rho_i),y_i) - \frac{1}{m}\sum_{i=1}^m f(\mathcal{Q}_{z_i}(x_i+\bar{\rho}_i),y_i)\right| \tag{31}$$

$$=\frac{2}{m}\mathbb{E}_Z\mathbb{E}_\Gamma\mathbb{E}_{\bar{\Gamma}}\mathbb{E}_\Sigma \left| \sum_{i=1}^m \sigma_i \left( f(\mathcal{Q}_{z_i}(x_i+\rho_i),y_i) - f(\mathcal{Q}_{z_i}(x_i+\bar{\rho}_i),y_i)\right)\right| \tag{32}$$

$$\leq \frac{2}{m}\mathbb{E}_Z\mathbb{E}_\Gamma\mathbb{E}_{\bar{\Gamma}}\sqrt{\sum_{i=1}^m |f(\mathcal{Q}_{z_i}(x_i+\rho_i),y_i) - f(\mathcal{Q}_{z_i}(x_i+\bar{\rho}_i),y_i)|^2} \tag{33}$$

$$\leq \frac{2}{m}\mathbb{E}_Z\mathbb{E}_\Gamma\mathbb{E}_{\bar{\Gamma}}\sqrt{\sum_{i=1}^m \beta^2\|\mathcal{Q}_{z_i}(x_i+\rho_i) - \mathcal{Q}_{z_i}(x_i+\bar{\rho}_i)\|^2} \tag{34}$$

$$\leq \frac{2\beta}{m}\mathbb{E}_Z\sqrt{\mathbb{E}_\Gamma\mathbb{E}_{\bar{\Gamma}}\left[\sum_{i=1}^m \|\mathcal{Q}_{z_i}(x_i+\rho_i) - \mathcal{Q}_{z_i}(x_i+\bar{\rho}_i)\|^2\right]} \tag{35}$$

$$=\frac{2\beta}{m}\mathbb{E}_Z\sqrt{\sum_{i=1}^m \mathbb{E}_\rho\mathbb{E}_{\bar{\rho}}\|\mathcal{Q}_{z_i}(x_i+\rho) - \mathcal{Q}_{z_i}(x_i+\bar{\rho})\|^2} \tag{36}$$

$$=\frac{2\beta}{m}\mathbb{E}_Z\sqrt{\sum_{i=1}^m \gamma(x_i,y_i)} \tag{37}$$

$$\leq \frac{2\beta}{m}\sqrt{\mathbb{E}_Z\left[\sum_{i=1}^m \gamma(x_i,y_i)\right]} \tag{38}$$

$$=\frac{2\beta}{m}\sqrt{\sum_{i=1}^m \mathbb{E}_{z_i}\gamma(x_i,y_i)} \tag{39}$$

$$=\frac{2\beta}{\sqrt{m}}\sqrt{\mathbb{E}_z\gamma(x,y)} \tag{40}$$

Again, we apply Jensen's inequality to get (30). In (32), we introduce Rademacher variables $\Sigma := (\sigma_1,\cdots,\sigma_m)$ (i.e., each random variable $\sigma_i$ takes values in $\{-1,+1\}$ independently with equal probability 0.5). The Rademacher variables introduces a random exchange of the corresponding difference term. Since $\Gamma$ and $\hat{\Gamma}$ are independently sampled from the same distribution, such a swap gives an equally likely configuration. Therefore, the equality (32) holds. The inequality (33) is given by Khintchine's inequality. The inequality (34) makes use of the lipschitz condition of $f$. (35) is derived from Jensen's inequality and due to that square root is a concave function. (37) is by the definition of the local dispersion of $\mathcal{Q}_z$. Again, we apply Jensen's inequality to obtain (38). Equation (39) and (40) follow from the settings that each $z_i = (x_i, y_i)$ is i.i.d.

For the term ②, we have

$$\mathbb{E}_Z \mathbb{E}_{\hat{Z}} \left| \mathbb{E}_{\bar{\Gamma}} F(Z, \bar{\Gamma}) - \mathbb{E}_{\tilde{\Gamma}} F(\hat{Z}, \tilde{\Gamma}) \right| \tag{41}$$

$$= \mathbb{E}_Z \mathbb{E}_{\hat{Z}} \left| \mathbb{E}_{\bar{\Gamma}} \left[ \frac{1}{m} \sum_{i=1}^m f(\mathcal{Q}_{z_i}(x_i + \bar{\rho}_i), y_i) \right] - \mathbb{E}_{\tilde{\Gamma}} \left[ \frac{1}{m} \sum_{i=1}^m f(\mathcal{Q}_{\hat{z}_i}(\hat{x}_i + \tilde{\rho}_i), \hat{y}_i) \right] \right| \tag{42}$$

$$= \mathbb{E}_Z \mathbb{E}_{\hat{Z}} \left| \frac{1}{m} \sum_{i=1}^m \mathbb{E}_{\bar{\rho}_i} \left[ f(\mathcal{Q}_{z_i}(x_i + \bar{\rho}_i), y_i) \right] - \frac{1}{m} \sum_{i=1}^m \mathbb{E}_{\tilde{\rho}_i} \left[ f(\mathcal{Q}_{\hat{z}_i}(\hat{x}_i + \tilde{\rho}_i), \hat{y}_i) \right] \right| \tag{43}$$

$$= \mathbb{E}_Z \mathbb{E}_{\hat{Z}} \left| \frac{1}{m} \sum_{i=1}^m \mathbb{E}_{\rho} \left[ f(\mathcal{Q}_{z_i}(x_i + \rho), y_i) \right] - \frac{1}{m} \sum_{i=1}^m \mathbb{E}_{\rho} \left[ f(\mathcal{Q}_{\hat{z}_i}(\hat{x}_i + \rho), \hat{y}_i) \right] \right| \tag{44}$$

$$= \frac{1}{m} \mathbb{E}_Z \mathbb{E}_{\hat{Z}} \mathbb{E}_{\Sigma} \left| \sum_{i=1}^m \sigma_i \left( \mathbb{E}_{\rho} \left[ f(\mathcal{Q}_{z_i}(x_i + \rho), y_i) \right] - \mathbb{E}_{\rho} \left[ f(\mathcal{Q}_{\hat{z}_i}(\hat{x}_i + \rho), \hat{y}_i) \right] \right) \right| \tag{45}$$

$$\leq \frac{1}{m} \mathbb{E}_Z \mathbb{E}_{\hat{Z}} \sqrt{\sum_{i=1}^m \left| (\mathbb{E}_{\rho} \left[ f(\mathcal{Q}_{z_i}(x_i + \rho), y_i) \right] - \mathbb{E}_{\rho} \left[ f(\mathcal{Q}_{\hat{z}_i}(\hat{x}_i + \rho), \hat{y}_i) \right]) \right|^2} \tag{46}$$

where equation (43) and (44) are due to each $\hat{\rho}_i$ and $\tilde{\rho}_i$ is i.i.d. Again, we introduce Rademacher variables at (45) and apply Khintchine's inequality to get (46). For the term $\left| (\mathbb{E}_{\rho} \left[ f(\mathcal{Q}_{z_i}(x_i + \rho), y_i) \right] - \mathbb{E}_{\rho} \left[ f(\mathcal{Q}_{\hat{z}_i}(\hat{x}_i + \rho), \hat{y}_i) \right]) \right|^2$, we have

$$\left| \mathbb{E}_{\rho} f(\mathcal{Q}_{z_i}(x_i + \rho), y_i) - \mathbb{E}_{\rho} f(\mathcal{Q}_{\hat{z}_i}(\hat{x}_i + \rho), \hat{y}_i) \right|^2 \tag{47}$$

$$\leq (|\mathbb{E}_{\rho} f(\mathcal{Q}_{z_i}(x_i + \rho), y_i)| + |\mathbb{E}_{\rho} f(\mathcal{Q}_{\hat{z}_i}(\hat{x}_i + \rho), \hat{y}_i)|)^2 \tag{48}$$

$$\leq 2 \left| \mathbb{E}_{\rho} f(\mathcal{Q}_{z_i}(x_i + \rho), y_i) \right|^2 + 2 \left| \mathbb{E}_{\rho} f(\mathcal{Q}_{\hat{z}_i}(\hat{x}_i + \rho), \hat{y}_i) \right|^2 \tag{49}$$

where inequality (49) is derived by the inequality $(a + b)^2 \leq 2(a^2 + b^2)$. Returning to (46), we then have

$$\frac{1}{m} \mathbb{E}_Z \mathbb{E}_{\hat{Z}} \sqrt{\sum_{i=1}^m \left| (\mathbb{E}_{\rho} \left[ f(\mathcal{Q}_{z_i}(x_i + \rho), y_i) \right] - \mathbb{E}_{\rho} \left[ f(\mathcal{Q}_{\hat{z}_i}(\hat{x}_i + \rho), \hat{y}_i) \right]) \right|^2}$$

$$\leq \frac{1}{m} \mathbb{E}_Z \mathbb{E}_{\hat{Z}} \sqrt{\sum_{i=1}^m 2 \left| \mathbb{E}_{\rho} f(\mathcal{Q}_{z_i}(x_i + \rho), y_i) \right|^2 + \sum_{i=1}^m 2 \left| \mathbb{E}_{\rho} f(\mathcal{Q}_{\hat{z}_i}(\hat{x}_i + \rho), \hat{y}_i) \right|^2} \tag{50}$$

$$\leq \frac{1}{m} \sqrt{\mathbb{E}_Z \mathbb{E}_{\hat{Z}} \left[ \sum_{i=1}^m 2 \left| \mathbb{E}_{\rho} f(\mathcal{Q}_{z_i}(x_i + \rho), y_i) \right|^2 + \sum_{i=1}^m 2 \left| \mathbb{E}_{\rho} f(\mathcal{Q}_{\hat{z}_i}(\hat{x}_i + \rho), \hat{y}_i) \right|^2 \right]} \tag{51}$$

$$= \frac{1}{m} \sqrt{\sum_{i=1}^m 2 \mathbb{E}_{z_i} \left| \mathbb{E}_{\rho} f(\mathcal{Q}_{z_i}(x_i + \rho), y_i) \right|^2 + \sum_{i=1}^m 2 \mathbb{E}_{\hat{z}_i} \left| \mathbb{E}_{\rho} f(\mathcal{Q}_{\hat{z}_i}(\hat{x}_i + \rho), \hat{y}_i) \right|^2} \tag{52}$$

$$= \frac{2}{\sqrt{m}} \sqrt{\mathbb{E}_z \left| \mathbb{E}_{\rho} f(\mathcal{Q}_z(x + \rho), y) \right|^2} \tag{53}$$

$$\leq \frac{2}{\sqrt{m}} \sqrt{\sup_{z \in \text{supp}(\mathcal{D})} \left| \mathbb{E}_{\rho} f(\mathcal{Q}_z(x + \rho), y) \right|^2} \tag{54}$$

$$= \frac{2A}{\sqrt{m}} \tag{55}$$

The final line is derived by the condition that $\sup_{z \in \text{supp}(\mathcal{D})} |\mathbb{E}_{\rho} f(\mathcal{Q}_z(x + \rho), y)| = A$. This gives the final result

$$\mathbb{E}_U g(u_1 \cdots u_m) \leq \frac{2\beta}{\sqrt{m}} \sqrt{\mathbb{E}_z \gamma(x, y)} + \frac{2A}{\sqrt{m}}$$

Plugging back to (20), we derive the bound in Theorem 1. This completes the proof. $\qquad \square$

Lastly we want to remark that the bound is not trivial, since we have

$$
\begin{aligned}
A &= \sup_{z \in \operatorname{supp}(\mathcal{D})} |\mathbb{E}_\rho f(\mathcal{Q}_z(x+\rho), y)| \\
&\leq \sup_{z \in \operatorname{supp}(\mathcal{D})} \mathbb{E}_\rho |f(\mathcal{Q}_z(x+\rho), y)| \\
&\leq \sup_{z \in \mathcal{X} \times \mathcal{Y}} \sup_{\|\rho\|_\infty \leq \epsilon} |f(\mathcal{Q}_z(x+\rho), y)| \\
&\leq \sup_{v,y \in \mathcal{X} \times \mathcal{Y}} |f(v, y)| = B
\end{aligned}
$$

In fact, $A$ could be much smaller than $B$, meaning the bound is tight.

## C  LOCAL DISPERSION RESULTS ACROSS OTHER DATASETS

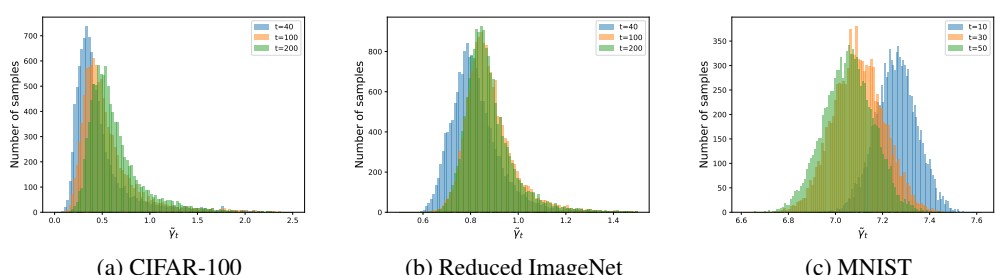

(a) CIFAR-100  (b) Reduced ImageNet  (c) MNIST

Figure 6: histograms of $\tilde{\gamma}_t$ on the CIFAR-100, Reduced ImageNet and MNIST testing set. On both CIFAR-100 and the Reduced ImageNet, the mode of the histogram shifts towards a larger number, indicating the level of $\tilde{\gamma}_t$ increases along adversarial training. By sharp contrast, on MNIST, the mode of the histogram shifts toward a smaller value. This behaviour matches the IDE results and the generalization bound derived in Theorem 1.

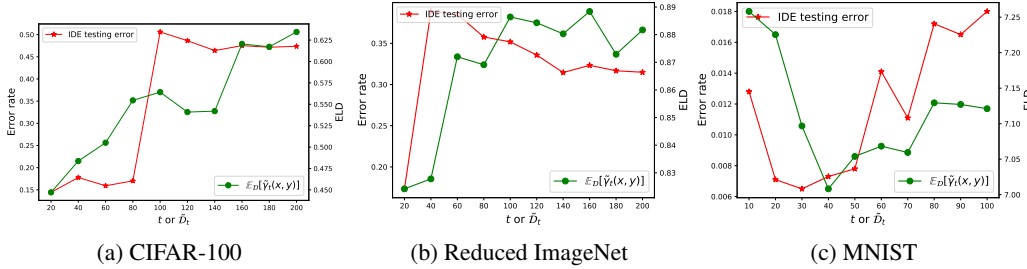

(a) CIFAR-100  (b) Reduced ImageNet  (c) MNIST

Figure 7: The expectation $\mathbb{E}_\mathcal{D}\tilde{\gamma}_t(x, y)$ evaluated on the CIFAR-100, Reduced ImageNet and MNIST testing set (green curves) and the corresponding IDE results (red curves). In each figure, the green and red curves are tightly correlated. This provides additional support to the conclusion in Theorem 1.

## D  ADDITIONAL RESULTS FOR SECTION 6

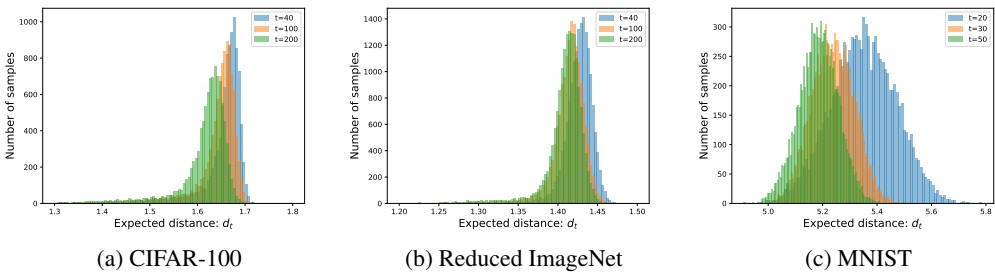

Figure 8: histograms of $d_t$ on the CIFAR-100, Reduced ImageNet and MNIST testing set. The reduction in the level of $d_t$ along adversarial training is shown in the figures.

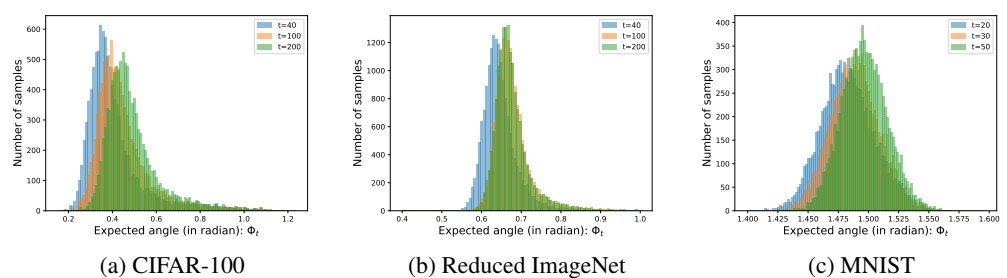

Figure 9: histograms of $\Phi_t$ on the CIFAR-100, Reduced ImageNet and MNIST testing set. It shows an increment in the level of $\Phi_t$ along adversarial training.

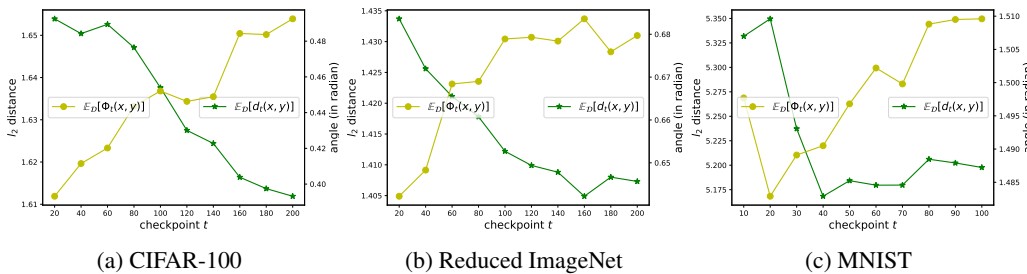

Figure 10: The evolution of $\mathbb{E}_{\mathcal{D}} d_t(x, y)$ and $\mathbb{E}_{\mathcal{D}} \Phi_t(x, y)$ along adversarial training evaluated on the testing set of CIFAR-100, Reduced ImageNet and MNIST. The behaviours of the two quantities are similar with those on CIFAR-10.

