# OpenReview forum: "On robust overfitting: adversarial training induced distribution matters"
_ICLR.cc/2024/Conference — Submitted to ICLR 2024_

### Official Review · Reviewer_Wjrq · 2023-10-25

**Soundness:** 1 poor
**Presentation:** 2 fair
**Contribution:** 1 poor
**Rating:** 3
**Confidence:** 4

**Summary:**

This paper aims to understand robust overfitting based on the evolution of the data distribution induced by adversarial perturbation along the adversarial training (AT) trajectory.
Specifically, the authors empirically find that such an induced data distribution becomes more and more difficult for a model trained (via standard training, not AT) on it to generalize as the AT continues.
A generalization upper bound is then proved for models obtained on the induced data distribution, which indicates that the generalization ability of those models is related to a newly proposed "local dispersion" term.
The authors then empirically demonstrate a correlation between the evolution of AT and the local dispersion.

**Strengths:**

1. The paper is clearly written and easy to follow.

2. The paper finds a new phenomenon that the data distribution induced by adversarial perturbation along AT is difficult for an ML model to generalize.

**Weaknesses:**

1. This paper seems trying to persuade readers that the proposed "local dispersion" plays an important role (even could be the cause) in robust overfitting, which however is misleading. Actually, this paper only EMPIRICALLY showed that there seems to be a correlation between local dispersion and robust overfitting, but unfortunately neither provided any theoretical explanation about the correlation nor explained how one can understand robust overfitting through this correlation. From this perspective, I would not treat "local dispersion" as a "contribution" of this paper.

2. The authors put a lot of effort into theoretically and empirically explaining the "generalization difficulty of the induced distribution", which however is weird. According to the paper itself, this "generalization difficulty" is only a result of robust overfitting, but could not be used to explain robust overfitting itself. In other words, the "generalization difficulty" as well as the "induced distribution" do not matter to robust overfitting (although the authors said they matter according to the title of this paper).

3. The contribution of this paper is very limited. As explained before, the only contribution of this paper is revealing the phenomenon that the data distribution induced by adversarial perturbation is difficult for an ML model to generalize. However, the authors did not show how one can leverage this phenomenon to explain AT, robust overfitting, or other aspects of machine learning. So I think the contribution of this paper is far below the standard of ICLR.

4. The generalization bound in Theorem 1 is misleading. Specifically, the authors replaced the term $\sup\_{x,y,\in\mathrm{supp}(\mathcal D)} | \mathbb{E}\_{\rho} f(\mathcal Q\_{x,y,\theta}(x+\rho),y) |$ with an assumed upper bound $A$. However, this replaced term also depends on the induced distribution $\mathcal Q\_{x,y,\theta}$, just like the local dispersion term. As a result, I think such a replacement (only replacing the term but not with the local dispersion term) is misleading and not appropriate.

**Questions:**

None.

---

> ### Author Response · Authors · 2023-11-22
> **Response to your comments**
>
> Thank you for your review. Towards the comments about the weakness of our paper, we would like to discuss and further explain it.
>
> 1. The observation that local dispersiveness correlates with robust overfitting is for the first time reported. In this respect, we believe it is a contribution. The bound provided in our Theorem 1 provides an theoretical explaination for understanding robust overfitting through local dispersion. Specifically, the bound suggests that local dispersiveness of the perturbation operators makes the induced distribution harder to learn, a phenomenon correlated well with robust overfitting.
>
> 2. We did observe correlation between generalization difficulty of the induced distribution and robust overfitting. However, based on this correlation, claiming "robust overfitting causes geneneralzation difficulty" or claiming the oppositve direction of causality are equally ungrounded.
>
> 3. We have not made any effort explain other aspects of machine learning beyond adversarial training. This however does not mean this work would not shed light beyond the context of this paper. The notion of generalization difficulty of a function class with respect to a distribution is an intriguing subject in its own right. In the context of iterative learning algorithms, we believe that this paper is in fact the first to touch on this topic. We disagree that "one can leverage this phenomenon to explain AT, robust overfitting". This entire paper is about leveraging this phenomenon to explain robust overfitting in PGD-AT.
>
> 4. It is not clear to us why the reviewer considers this treatment "misleading" and "not appropriate". Replacing a term with its upper bound is commonly used in deriving learning bounds. Whether it is appropriate depends on if it has made the bound too loose to characterize the studied behaviour of interest. In our bound, treating $A$ as a constant still allows the bound to correctly reflect the generalization behaviour in IDE experiments. Then, there is no basis to regard our treatment inappropriate.

---

> ### Comment · Reviewer_Wjrq · 2023-12-04
> **I vote for rejection**
>
> Thanks to the authors for their very timely rebuttal. Unfortunately, the rebuttal resolves none of my concerns and there left no time for further discussion with the authors since the authors submitted their rebuttal in the last day of discussion period.
> As a result, I keep my score and vote for rejection.
>
> Detailed comments are as follows:
> - To Q1: Your "finding" is foreseeable and trivial. It is mainly due to the fact that the distribution of adversarial examples is different from the distribution of clean examples, which however is obvious.
>
> - To Q2: I agree with Reviewer 3NwP that correlation does not imply causality. Please refer to the example in Reviewer 3NwP's response for details.
>
> - To Q3: Again, your paper is not trying to explain robust overfitting. It only analyzes a trivial phenomenon caused by AT and this phenomenon is not the reason of causing robust overfitting. Please remind that even in your paper there is no theoretical analysis supporting the claim that "local dspersion cause robust overfitting".
>
> - To Q4: My concern is that since both $\sup\_{x,y,\in\mathrm{supp}(\mathcal D)} | \mathbb{E}\_{\rho} f(\mathcal Q\_{x,y,\theta}(x+\rho),y) |$ and $\mathbb{E}\_{\mathcal D} \hat\gamma\_\theta(x,y)$ depend on $\mathcal Q\_{x,y,\theta}$, only bounding the first term with a constant but not the last term is inappropriate. It is possible that the scale of $\sup\_{x,y,\in\mathrm{supp}(\mathcal D)} | \mathbb{E}\_{\rho} f(\mathcal Q\_{x,y,\theta}(x+\rho),y) |$ is much larger than the scale of $\mathbb{E}\_{\mathcal D} \hat\gamma\_\theta(x,y)$, which thus results in $\sup\_{x,y,\in\mathrm{supp}(\mathcal D)} | \mathbb{E}\_{\rho} f(\mathcal Q\_{x,y,\theta}(x+\rho),y) |$ dominating the upper bound in Theorem 1 and the influence of $\mathbb{E}\_{\mathcal D} \hat\gamma\_\theta(x,y)$ becoming negligible.

---

### Official Review · Reviewer_3NwP · 2023-10-31

**Soundness:** 2 fair
**Presentation:** 2 fair
**Contribution:** 2 fair
**Rating:** 3
**Confidence:** 4

**Summary:**

This paper study an important phenomenon in adversarial training, which is called robust overfitting. The author shows that robust overfitting can be attributed to the (adversarial) induced distribution and local dispersion.

**Strengths:**

This paper study an important phenomenon in adversarial training, which is called robust overfitting. The author provides an observation that the following two generalization gaps looks similar:

1) robust generalization gap of adversarial training
2) standard generalization gap of standard training on the induce destruction in step t, $\tilde{D}_t$.

This observation provides a good interpretation of robust overfitting and provides a way to understand robust overfitting by studying the evolution of $\tilde{D}_t$.

The authors further provide a generalization analysis w.r.t.  $\tilde{D}_t$, based on the uniform convergence bound.

**Weaknesses:**

This paper attributes robust overfitting to the 'local dispersion' represented by $\tilde{\gamma}_\theta(x,y)$. An important factor in this quantity is the initialization of PGD attacks. However, the analysis appears to falter in a basic sanity check: zero-initialization PGD-adversarial training. Specifically, we examine a simplify PGD attack that always starts from the clean sample. From my observations (if I correct), such a PGD attack is already powerful, and robust overfitting appears even in this PGD-adversarial training scenario.

In this scenario, $\tilde{\gamma}_\theta(x,y)=0$, leaving only the last two terms in the upper bound as per Theorem 1. Consequently, local dispersion is not related to robust overfitting. In my opinion, the uniform convergence bounds are overly large, and the first term may not be directly related to the generalization gap. Since the proof is based on a modification of Rademacher analysis, it would be better to show that the bound in Theorem 1 is tighter than a simpler, for example, Rademacher complexity bound. It seems that the proposed bound is not tighter by comparing $2A$ (proposed bound) with $2\beta B_x$ (RC bound).

In contrast, if I my observation is wrong, it would be better if the authors provide experiments to dismiss it.

Therefore, attributing robust overfitting to local dispersion is not very convincing to me.

I believe this paper has merit but isn't ready for publication. While Section 4 is insightful, Section 5 requires revisions. I recommend the authors address these concerns. I'm willing to engage in a discussion with the authors and am flexible about adjusting my score.

Minor suggestion:

In Figure 1 and 2, it would be better to directly show the trends of IDE test error v.s. robust test error, or  IDE generalization gap v.s. robust generalization gap.

**Questions:**

See weakness.

---

> ### Author Response · Authors · 2023-11-22
> **Reply to your questions (part 1)**
>
> We would like to sincerely thank the reviewer for your sharp questions, particularly the one pointing to the fact that PGD-AT with zero-initialized PGD attacks may also lead to robust overfitting. After seeing your comments, we performed some experiments in this direction and indeed observe a similar robust overfitting phenomenon.
>
> This has made us rethink what should the correct formalism for our development. In particular, the experimental results we observe in Section 4 made us firmly believe that local dispersion is still the correct measure, extremely relevant to robust overfitting (we did try a few other local measures, for which generalization bounds can be derived for explaining the IDE experiments, but none of those measures when evaluated empirically shows a strong correlation).
>
> Our key resolution is to view the dataset (say CIFAR10) as being drawn from a distribution obtained by randomly perturbing a more robust distribution. Specifically, denote this more robust distribution by ${\cal D}^*$, and it satisfies the property that when $x$ is in the support of ${\cal D}^*_{\cal X}$ (the marginal of ${\cal D}^*$ on the input space), all points in the $\epsilon$-ball around $x$ have the same label as $x$. The data distribution ${\cal D}$ is obtained by the following generative process: draw $(x, y)$ from ${\cal D}^*$ and create $(x+\rho, y)$ with $\rho$ drawn from the uniform distribution on the infinity-norm ball ${\mathbb B}(0, \epsilon)$), and regard the distribution of $(x+\rho, y)$ as ${\cal D}$.
>
> This formalism may appear counter-intuitive, since the usual perspective is that the data distribution ${\cal D}$ by itself is "robust", but here our formalism insists that ${\cal D}^*$ is perfectly $\epsilon$-robust while ${\cal D}$ may not be.  However, we argue that we never know if the data distribution ${\cal D}$ is truly robust. The best one can claim is that the **dataset** drawn from ${\cal D}$ appears robust. It might be possible that the points living in the low-probability region of ${\cal D}$ are in fact never seen in the dataset, and such points are not robust. On the other hand, the fact that the dataset is robust can be well argued within our formalism. In our formalism, the support ${\rm supp}({\cal D} _ {\cal X})$ of ${\cal D} _ {\cal X}$ and the support ${\rm supp}({\cal D}^* _ {\cal X})$ of ${\cal D} _ {\cal X}^*$ differ only slightly (by an $\epsilon$-wide boundary of ${\cal D} _ {\cal X}$), then the density of ${\cal D}^* _ {\cal X}$ covers almost all probability under ${\cal D} _ {\cal X}$. As a consequence, when one draws a dataset from ${\cal D}$, with high probability, all points in the dataset fall in the support of ${\cal D}^* _ {\cal X}$ and hence are robust by definition.
>
> We have rewritten some of the paragraphs to illustrate this formalism in Section 5. The statement of Theorem 1 is also slightly modified, where the expectated local dispersion (ELD) is now defined using expectation over ${\cal D}^*$. The estimation of ELD is still using average over the dataset, since using the argument above (and also a similar argument in the blue text above Figure 4), such a dataset can also be regarded as being drawn from ${\cal D}^*$.
>
> We believe that this reformulation address your concern centered around zero-initialized PGD attacks: even when in PGD-AT, no random initialization is applied to the data points, each data point (drawn from ${\cal D}$) is already perturbed with random noise, by definition.
>
> We wish to note that because our effort and time spent on resolving your question, we were not able to submit our response in a timely manner and perhaps have missed the opportunity of further discussion with you. We hope that we have adequately addressed this concern of yours. Any input is nonetheless welcome.

---

> ### Author Response · Authors · 2023-11-22
> **Reply to your questions (part 2)**
>
> **Compare with the Rademacher complexity (RC) bound**
>
> Regarding your comment of comparing with RC bounds, we made an effort along this direction. It is however unfortunate that with only liptchitz condition of $f$ alone, it is difficult to obtain a meaningful RC upper bound. Below we show our derivation steps. Any comments are welcome.
>
> Using the RC bound involves analysing the RC of the hypothesis set ${\cal F}$ w.r.t the data distribution $\tilde{\mathcal{D}}_{\theta}$. defined as
> $\mathcal{R} _ {\tilde{\mathcal{D}} _ {\theta}}({\cal F}):= \frac{2}{m}{\mathbb E} _ {\tilde{\mathcal{D}} _ {\theta}}\mathbb{E} _ {\Sigma}\sup\limits _ {f\in{\cal F}}|\sum _ {i=1}^{m}\sigma  _i f(v_i,y_i)|$
>
> where $\Sigma:=(\sigma_1,\cdots ,\sigma_m)$ denotes $m$ indepent Rademacher variables and $\{(v_i,y_i)\}$ is an i.i.d sample from the distribution  $\tilde{\mathcal{D}} _ {\theta}$. By the construction of $\tilde{\mathcal{D}}_{\theta}$, we further write $\mathcal{R} _ {\tilde{\mathcal{D}} _ {\theta}}({\cal F})$ as
>
> $\mathcal{R} _ {\tilde{\mathcal{D}} _ {\theta}}({\cal F}):= \frac{2}{m}{\mathbb E} _ {\mathcal{D}^{*}}{\mathbb E} _ {\Gamma}\mathbb{E} _ {\Sigma}\sup\limits _ {f\in{\cal F}}|\sum _ {i=1}^{m}\sigma_i f({\mathcal Q} _ {x_i, y_i, \theta}(x_i+\rho_i),y_i)|$
>
> where $\Gamma:=(\rho_1,\cdots, \rho_m)$ with $\rho_i$'s drawn i.i.d from $\mathcal{U}([-\epsilon, +\epsilon]^{d})$.  This quantity can be further factorized by adding and substracting the term $\sigma_i f(x_i+\rho_i,y_i)$ and applying the triangle inequality:
>
> $\mathcal{R} _ {\tilde{\mathcal{D}} _  {\theta}}({\cal F}):= \frac{2}{m}{\mathbb E} _ {\mathcal{D}^{*}}{\mathbb E} _ {\Gamma} \mathbb{E} _ {\Sigma}\sup\limits _ {f\in{\cal F}}|\sum _ {i=1}^{m} \sigma _ i \left(f({\mathcal Q} _ {x _ i, y _ i, \theta}(x _ i+\rho _ i), y _ i)- f(x _ i+\rho _ i,y _ i)\right) +  \sigma _ i  f(x _ i+\rho _ i, y _ i)|$
>
> $\le \frac{2}{m}{\mathbb E} _ {\mathcal{D}^{*}}{\mathbb E} _ {\Gamma}\mathbb{E} _ {\Sigma}\sup\limits _ {f\in{\cal F}}|\sum _ {i=1}^{m}\sigma _ i \left(f({\mathcal Q} _ {x _ i, y _ i, \theta}(x _ i+\rho _ i) , y _ i)- f(x _ i+\rho _ i, y _ i)\right)| $
>
> $+  \frac{2}{m}{\mathbb E} _  {\mathcal{D}^{*}}{\mathbb E} _ {\Gamma}\mathbb{E} _ {\Sigma}\sup\limits _ {f\in{\cal F}}|\sigma _ i f(x _ i+\rho _ i, y _i )|$
>
> $\le \frac{2}{m}{\mathbb E} _ {\mathcal{D}^{*}}{\mathbb E} _ {\Gamma}\mathbb{E} _ {\Sigma}\sup\limits _ {f\in{\cal F}}|\sum _ {i=1}^{m}\sigma _ i \left(f({\mathcal Q} _ {x _ i, y _ i, \theta}(x _ i+\rho _ i) , y _ i)- f(x _ i+\rho _ i, y _ i)\right)| + \mathcal{R} _ {{\mathcal{D}}}({\cal F}) $
>
> where the second term in last line is the RC of ${\cal F}$ w.r.t the original data distribution ${\cal D}$. It characterizes the generalization behaviour of ${\cal F}$ on the original data distribution, which is out of our interest. We now use the lipchitz condition of $f\in{\cal F}$ to derive an upper bound for the first term.
>
> $ \frac{2}{m}{\mathbb E} _ {\mathcal{D}^{*}}{\mathbb E} _ {\Gamma}\mathbb{E} _ {\Sigma}\sup\limits _ {f\in{\cal F}}|\sum _ {i=1}^{m}\sigma _ i \left(f({\mathcal Q} _ {x _ i, y _ i, \theta}(x _ i+\rho _ i) , y _ i)- f(x _ i+\rho _ i, y _ i)\right)| \quad(1)$
>
> $\le \frac{2}{m}{\mathbb E} _ {\mathcal{D}^{*}}{\mathbb E} _ {\Gamma}\sum _ {i=1}^{m}\beta  || {\mathcal Q} _ {x _ i, y _ i, \theta}(x _ i+\rho _ i)- (x _ i+\rho _ i)  || _ 2 \quad(2)$
>
> $= 2\beta {\mathbb E} _ {\mathcal{D}^{*}}{\mathbb E} _ {\rho}||{\mathcal Q} _ {x, y, \theta}(x+\rho)- (x+\rho)|| _ 2 \quad(3)$
>
> Equation (2) above holds for the following reason.  For arbitrary fixed choice of $\{(x_i,y_i)\}_{i=1}^{m}$ , $(\rho_1,\cdots, \rho_m)$ and $(\sigma_1,\cdots ,\sigma_m)$, let's assume the supremum in (1) is achievable and can be achieved by some $f^{*}\in {\cal F}$. Then
>
> $\sup\limits_{f\in{\cal F}}|\sum_{i=1}^{m}\sigma_i \left(f({\mathcal Q}_{x_i, y_i, \theta}(x_i+\rho_i),y_i)- f(x_i+\rho_i,y_i)\right)|$
>
> $=|\sum_{i=1}^{m}\sigma_i \left(f^* ({\mathcal Q}_{x_i, y_i, \theta}(x_i+\rho_i),y_i)- f^* (x_i+\rho_i,y_i)\right)|$
>
> $\le \sum_{i=1}^{m} |\sigma_i \left(f^* ({\mathcal Q}_{x_i, y_i, \theta}(x_i+\rho_i),y_i)- f^* (x_i+\rho_i,y_i)\right)|$
>
> $\le\sum _ {i=1}^{m}\beta||{\mathcal Q} _ {x _ i, y _ i, \theta}(x _ i+\rho _ i)- (x _ i+\rho _ i)||_2\quad (4)$
>
> The inequality in (4) is due to that each $f\in {\cal F}$ is $\beta$-lipschitz. Note that this part of derivation hold for arbitary choice of  $\{(x_i,y_i)\}_{i=1}^{m}$ , $(\rho_1,\cdots, \rho_m)$ and $(\sigma_1,\cdots ,\sigma_m)$ . Thus we have the inequality hold in (2). It is unfortunate that the upper bound (3) does not decay with $m$.
>
> Here we've made an effort to compare our bound with the RC bound, however, it seems to be hard to directly derive a meaningful and non-vacuous upper bound for RC with only the lipschitz and boundedness conditions of $f$.

---

> > ### Comment · Reviewer_3NwP · 2023-11-23
> > **Thank you very much for the response**
> >
> > Thank you very much for the response.
> >
> > It seems that the reason why the authors "firmly believe that local dispersion is still the correct measure" is based on the experiments shown in figure 4.
> >
> > It is shown that "the level of LD (and ELD) increases during PGD-AT and correspondingly the model obtained from Dt becomes harder to generalize". However, it only shows that these two are positively related. There is a huge gap between positive relation and causality. Thus, the claim " in influencing the generalization" and " in affecting the generalization performance" in Sec. 5 is very strong to me.
> >
> > To achieve positive relation is easy. For example: One can plot another figure: the cost of money for training increases during PGD-AT and correspondingly the model obtained from Dt becomes harder to generalize. It implies the cost of money is positively related to robust overfitting. But robust overfitting is not due to the cost of money increases.
> >
> > Actually, the counter example is to show that they are not related in a different experiment, not to mention causality.

---

> ### Author Response · Authors · 2023-11-23
> **Causality versus Correlation**
>
> We fully agree that correlation does not mean causality. In fact, we had made the same remark in our response to another review.
>
> In the case of robust overfitting versus generalization difficulty of D_t,  the two factors are inter-leaved during PGD-AT, since D_t  depends on the current $\theta_t$, while directly impacting the update of $\theta_t$ to $\theta_{t+1}$. For this reason, also in light of our experimental observations, we hope you would agree that, between the two, there is a real  correlation (not a spurious one as you constructed), and that it is not too strong to say "the two factors influence each other".
>
> Whether or not our result is convincing to you, we would like to express our sincere gratitude for your questions. They certainly have helped us to improve the formalism of this work. That, by itself, has already made this submission worthwhile.

---

### Official Review · Reviewer_NYh7 · 2023-11-01

**Soundness:** 2 fair
**Presentation:** 3 good
**Contribution:** 1 poor
**Rating:** 5
**Confidence:** 4

**Summary:**

**Update:** I increased my score to "marginally below acceptance threshold" from "reject". I think the paper is not ready for publication at this stage, but it has the potential to evolve into a strong paper in the future.

Adversarial training, despite obtaining robustness against adversarial attacks, suffers from overfitting. Historically, this was found to be surprising given that adversarially trained models are supposed to be "robust"; however, over time, it became clear that robustness against adversarial attacks and robustness against overfitting are different concepts. Hence, recent work started to focus on developing adversarially robust models that do not overfit (or, to battle the so-called "robust overfitting"). To this end, it has been crucial to first understand the key reasons behind the overfitting of adversarially robust models. This work investigates this question specifically for the PGD-AT (adversarial) training of neural networks, which, in the stage of "nature's problem" applies projected gradient ascent to perturb training instances within the allowed "budget" of perturbations [PGD], and in the stage of "updating the model after the previous attack" applies one iteration of gradient descent where the gradients are with respect to the losses of the perturbed instances. The reason for the so-called "robust overfitting" is hypothesized to be the overfitting over the distribution that is induced by the "nature's problem" over the iterations of PGD-AT, which means that throughout the iterations of PGD-AT, we start to overfit the distribution that is a result of perturbing the true feature values. To this end, the authors first show that the generalization error of the original dataset and the generalization error of the shifted distributions by nature throughout the iterations have a similar pattern. Then, to formally back this claim, they derive a theorem that upper bounds the deviation of the empirical risk and the true risk by some intuitive parameters of nature's data-shifting mechanism in addition to the parameters associated with the loss function.

**Strengths:**

I believe the authors study a very relevant and modern problem. Understanding the sources of robust overfitting is essential for developing intelligence models that are robust against multiple sources of errors (attacks, data shifts, etc.). The motivation behind the paper is clear. The experiments are also using very recent practice, including how they make a Reduced ImageNet as the recent works.

**Weaknesses:**

I have several concerns about the paper. I will first provide a high-level summary of my major concerns, and then provide an in-depth discussion of those, followed by some milder concerns. My recommendation is a rejection, however, I will have a very open mind and stay active during the discussion period as I can see the authors put a lot of effort into this paper and I want to make sure I do not overlook anything.
-- --
**Summary of major concerns**
1. I do not find the key message "robust overfitting is a result of overfitting the adversarially induced data" to be surprising. This should already be clear. This paper re-discovers this well-known effect for the specific PGD-AT algorithm.
2. [Biggest concern] Not a single paper from distributionally robust optimization is cited. For years there have been many variants of Theorem 1, bounding the deviation of the empirical risk from the true risk under the assumption that the training set is sampled i.i.d. from a data-generating distribution. There are quite strong results out there. Not only do these works derive upper bounds on the risk deviations, but they also *do* develop and solve distributionally robust counterparts of adversarial training.
3. The discussions are hard to follow, most conclusions are heuristic rather than theoretical, and there are several vague statements.
-- --
**Major concern 1**

I think the reader could be misled when we say robust overfitting is not understood. Firstly, I believe the definition of overfitting is more accurate in the recent ICML paper "Certified Robust Neural Networks: Generalization and Corruption Resistance" by Bennouna et al., where the source of overfitting is also explained accurately (especially, please see Section 3).

I believe this paper (and most of the cited related work) is concerned with the structure of overfitting as a result of the iterations of a specific training algorithm. The structure of the PGD-AT algorithm, for example, drives the way the induced distribution behaves, as the nature attacks, optimization parameters are revised, and the nature attacks again upon seeing the previous parameters, and so on; so, the data distribution keeps deviating as we update our parameters. I would recommend a clear discussion of this.

Empirical risk minimization (ERM) minimizes $\underset{(x,y) \sim S}{\mathbb{E}}[l_{\theta}(x,y)]$ where $S$ is i.i.d. drawn from $\mathcal{D}$, and we know it overfits if we compare it with $\underset{(x,y) \sim \mathcal{D}}{\mathbb{E}}[l_{\theta}(x,y)]$. Keeping this in mind, note that when adversarial attacks happen, the true risk is not $\underset{(x,y) \sim \mathcal{D}}{\mathbb{E}}[l_{\theta}(x,y)]$ anymore, rather it is $\underset{(x,y) \sim \mathcal{D}}{\mathbb{E}}[\max_{v \in \mathbb{B}(x,\varepsilon)} l_{\theta}(v,y)]$. If we therefore define a new loss function $l^\varepsilon_{\theta} (x, y) :=  \max_{v \in \mathbb{B}(x,\varepsilon)} l_{\theta}(v,y)$, we can see that training $\underset{(x,y) \sim S}{\mathbb{E}}[l_{\theta}(x,y)]$ while testing $\underset{(x,y) \sim \mathcal{D}}{\mathbb{E}}[l^\varepsilon_{\theta}(x,y)]$ is inconsistent. Therefore, the adversarial training paradigm of Madry et al. (2019) instead trains $\underset{(x,y) \sim S}{\mathbb{E}}[l^\varepsilon_{\theta}(x,y)]$. Now, it must be clear that both training and test/true risks are expectations of $l^\varepsilon_\theta$ over $S$ and $\mathcal{D}$. Adversarial training is thus another ERM problem, with a different loss that incorporates the adversarial perturbations. So, the whole overfitting theory is applicable. This discussion also shows why the "induced distribution overfitting" is not interesting, as if we revise the training datasets wrt the perturbations of $l^\varepsilon$, then what this paper discovers is the classical overfitting. (Again, the paper cited above should provide a more accurate description). In other words, there is a 1:1 analogy between the overfitting while training $l^\varepsilon_\theta$ over $S$ versus keeping the loss as $l_\theta$ and instead perturbing the underlying datapoints and observing overfitting on this perturbed data.
-- --
**Major concern 2**

The previous discussion hints that robust overfitting, as the classic overfitting, is a result of the optimizer's curse, that is, we optimize the parameters over $S$ whereas it is i.i.d. sampled from a true data generating distribution $\mathcal{D}$ and there will be a nonzero distance between $S$ and $\mathcal{D}$. Hence, there is a lot of work that does distributionally robust (DR) training of models.

I think this paper talks about concepts extremely relevant to DRO (data shifts, overfitting, etc.). Definition 1, for example, appears to be a simplified version of the Wasserstein-Kantorovich-like duality we always use in DRO. Theorem 1 bounds the difference between the empirical and true risk, which is studied extensively in the DRO literature. Theorem 1 can already be tightened in my view with the techniques in "Wasserstein distributionally robust optimization: Theory and applications in machine learning" by Kuhn et al. (2019) especially since the loss function is assumed to be Lipschitz continuous (then with proofs similar to the paper I just cited you can ignore the term B). Simply, the left-hand side of the quantity in theorem 1 can be upper bounded by $\beta \cdot \varepsilon$ where $\varepsilon$ is the Wasserstein distance between $S$ and $\mathcal{D}$ -- in practice, $\varepsilon$ is not known, but it is not an issue here since Theorem 1 is 'w.p. at least $1 - \tau$' (that said, here probably the authors mean 'confidence') which means that with the same confidence, one can retrieve $\varepsilon$ via the finite sample guarantees (see, e.g., Theorem 3.4 of "Data-driven distributionally robust optimization using the Wasserstein metric: performance guarantees and tractable reformulations" by Mohajerin Esfahani and Kuhn, 2018).

Overall, I am concerned about the fact that DRO (and related fields) are not mentioned in this paper whereas the underlying goal is identical. Also, note that the DRO papers have stronger bounds than Theorem 1, and they also have algorithms to overcome the overfitting (or at least improve a little bit); hence I believe the contribution of this paper is limited.

-- --
**Major concern 3**

Please find a (rather unorganized) list of points that I believe make the paper hard to follow.
- In the abstract, the terms used will not be clear until one reads the paper. For example "evolution of the data distribution" will be defined later on and here it does not make meaning. Similarly "checkpoints", or "certain local parameters" are unclear yet. Also in the abstract, there is circular reasoning: initially, it is said this paper will explain why robust overfitting happens, but then "We observe that the obtained models become increasingly harder to generalize when robust overfitting occurs" -> Aren't we investigating the reason behind robust overfitting?
- "robust population risk $R_{\mathcal{D}}^{\text{rob}}$" is defined wrongly. This is adversarial risk. Robust refers to the model that is trained adversarially robust. The definition the authors provide is for any $\theta$, regardless of how they are trained. Overall, the paper uses "robust" and "adversarial" interchangeably, and this should be corrected.
- "As such, one may expect the generalization performance of a model trained on $\mathcal{D}_t$ is similar to that on $\mathcal{D}$" would contradict every paper that is concerned with distributional shifts and DRO.
- Some terms are used before they are defined or abbreviated. Examples: "PGD" name is being used in the beginning before it is defined in S3.
- There is some mystery behind discussions. In the beginning, for example, overfitting is said to be "unexpected" -> why?
- Some statements are informal. Examples: "A *great deal* of research effort has been spent" is not followed by any references. On the next page, "does robust overfitting have anything to do with this" is also informal.
- "In Dong et al. (2021a), the authors observe the existence of label noise in adversarial training" -> What is a label noise, could the authors please briefly clarify? That said, "in X et al. (...)" is the wrong format as "X et al." refers to authors rather than the paper -- maybe use `citep`? The paragraph later mentions "analysis of memorization" which is also not clear as we are talking about an algorithm. The sentence  with "small loss" can also be revised.
- Page 1, "smoothing logits" -> not clear. In general, when a paper is cited, please make sure to not assume the reader knows the specific terminology.
- Section 2, after Figure 1 has "Having established generalization difficulty of the induced distribution". There are two issues here; (i) the sentence is really hard to grasp, what is the "generalization difficulty of the induced distribution"? (ii) after showing an experimental plot, the authors claimed something is established, but it is not. This pattern appears throughout the paper, plots or experiments are used as evidence for theoretical significance, but I find these not convincing. (Similarly, the second sentence of Section 5 starts with "This suggests".)
- The paragraph at the end of page 2: Again, this is specific to the training method.
- "The most popular adversarial training" is said to be PGD-AT, but this is specific to a class of learners, e.g., neural networks.
- Could the authors cite some generic paper for the PGD-AT as described from the end of page 3 onward?
- The described PGD-AT provides an approximate solution, not necessarily a strong solution. However, the conclusions drawn in this paper are for adversarial learning in general. This distinction could be clarified.
- Section 6 is overall a heuristic approach, and informal. If the preceding sections were formal enough, this would be a good teaser, but at the current form, I think it makes the paper less formal.

-- --
**Minor (typos, etc.)**
- "may indeed correlates"
- "few percent" on page 2 is vague.
- "dispersion property" does not have a meaning at the beginning of the paper until it is defined at the end.
- "dispersivenss" could be a typo.
- "perturbation decreases it magnitudes" has typos.
- Title of Section 2 "Other Related Works" could simply be "Related Work"
- "Gaussian and Bernuolli model" -> model's'
- The "paper" -> the work
- "an one-step PGD" -> a
- $\mathrm{sgn}$ is a *vector* of signs, please clarify somewhere.
- Page 7, before "More specifically" there is a missing period.
- In several places the quotes start with the wrong direction. Perhaps use `` '' in LaTeX instead of " ".
- "CIFAR 100(Krizhevsky et al., 2009)" -> missing space
- extra space after "i.i.d." can be prevented by '\@' in LaTeX

**Questions:**

- Introduction: "error probability in the predicted label for adversarially perturbed instances" -> Is "probability" the correct word?
- For clarity, can the authors specify what each "error" looks like, maybe via mathematical notation? In the above sentence, for example, "the error in the predicted label for adversarially perturbed instances" is mentioned, but the structure of adversarial attacks comes later. That said, I was also wondering, can the authors add a sentence discussing what part of their proofs would not work in the case of $\ell_{p  \neq \infty}$ norm attacks? Finally, the $\ell_p$ attacks, even when they are not restricted to $\infty$-norms, typically deserve a quick discussion on why such attacks are interesting.
- Page 2 says Dong et al and Yu et al. conflict 'to' each other (conflict "with" might perhaps be the better word): could the authors please explain a little bit about how these explanations vary? The sentence that follows says "each shown to" which probably is a typo.
- Before the PGD training is mentioned, the paper uses the phrase "along adversarial training". I think the "along" implies that the algorithm is iterative and again it is specific to the structure of the PGD-AT algorithm. Could the authors please mention this at the beginning of the paper? Similarly, after that "induces a new data distribution" is not immediately clear; I only understood what is meant here after I read the paper fully once. Please also clarify "at training step $t$" terminology.
- Right before Figure 1, could the authors please include the framework of IDE testing step by step? It does not need to be long but it would improve the readability. I understood the whole story much better in the first paragraph of Section 4.
- Can the authors please cite WRN-34 and PRN-18 when they first appear?
- End of Section 2: can you please elaborate on the "mismatch between theoretical settings and real-world practices"
- Could the authors please clarify what on page 4 "**distribution** of $(v,y) = $..." refers to? Is it the distribution whose randomness comes from $\rho$? Or do the authors mean the collection of training instances perturbed with respect to the underlying mechanism?
- I guess the name "checkpoints" is coined because it is expensive to do this IDE check at every iteration. Would it perhaps be less confusing if we do not mention checkpoints, rather say "IDE is recorded every $k$ iterations"?
- The experiments are repeated only 5 times, isn't this also contradicting the stochastic optimization literature/curse of dimensionality etc., as the choice of $\rho$ is crucial? Maybe the SAA literature has an answer to this. (Similarly the 10 pairs of $\rho, \rho'$ need discussion).
- Why are the parameters not CV'ed? It is ok to just cite a paper using similar combinations.
- Can Figure 3 (a) be plotted over the error rate range [0, 0.1]? It is hard to read on this scale.
- Is there a reason why Section 5 uses $f \in \mathcal{F}$ notation rather than keeping the adversarial attacks explicit? Would the proof make sense if $\varepsilon = 0$ as well? Also, the $f \in \mathcal{F}$ refers to **any** function, can the authors make it clear how the adversarial setting is used in the proof? I guess the only property of $f$ about the adversarial training is the "boundedness of perturbation-smoothed loss) -- isn't this very loose?
- "measurable" -> With respect to which algebra? (I am asking because the norm space is explicitly defined).

---

> ### Author Response · Authors · 2023-11-22
> **Reply to major concern 1**
>
> Thank you very much for your careful review.  We believe that the reviewer has misunderstood the main point of our work.
>
> It is correct, as you stated, that "robust overfitting is a result of overfitting the adversarially induced data" (we will discuss this statement in detail later). It is also correct that one may consider adversarial training as training with a modified loss function and apply standard learning-theoretic analysis to analyze the generalization behaviour of adversarial training. However, such a view is fundamentally limited, since it fails to explain why adversarily trained model is much more prone to overfitting, for example on CIFAR10, than models trained on the clean dataset (using the standard loss). This paper studies the generalization behaviour in adversarial training by zooming into the dynamics of PGD-AT iterations and attempts to understand the cause leading to the final induced distribution difficult to generalize. Such a philosophy, i.e., tracking the training dynamics and understanding its impact on the generalization behaviour of the learned model, has demonstrated great successes in deep learning research, for example, in Arora et al, "Fine-Grained Analysis of Optimization and Generalization for Overparameterized Two-Layer Neural Networks",  in Soudry et al, "The Implicit Bias of Gradient Descent on Separable Data",  in Lyu et al, "Gradient Descent Maximizes the Margin of Homogeneous Neural Networks" and in various works on Neural Tangent Kernel (Jacot, 2018). Such algorithm-dependent analysis has proven to be much more relevant to deep learning models than the pessimistic bounds given by classical learning theory, which puts primary emphasis on the loss function and the hypothesis class.
>
> The reviewer considers that the key message of our paper is "robust overfitting is a result of overfitting the adversarially induced data" and that we merely "have rediscovered" this. We like to note: 1) this is not our key message, 2) the presented results are much more than rediscover this.  First, when speaking of "robust overfitting is a result of overfitting the adversarially induced data", the adversarially induced data (distribution) refers to the distribution induced by adversarial perturbation against the final learned model, or, using the notation of this paper, it refers to the distribution $\tilde{{\cal D}}_T$ at final iteration $T$. Empirical results have shown that the learned model from PGD-AT does not generalize well with respect to $\tilde{{\cal D}}_T$, hence the conclusion "robust overfitting is a result of overfitting the adversarially induced data". This phenomenon requires no re-discovery. The main question motivating this research is why $\tilde{{\cal D}}_T$ becomes difficult to generalize. We thus set out to investigate how the perturbation-induced distributions $\tilde{{\cal D}}_t$ evolve along the trajectory of PGD-AT and the factors that may cause $\tilde{{\cal D}}_t$ to be increasingly difficult to generalize.

---

> > ### Author Response · Authors · 2023-11-22
> > **Reply to major concern 1 (Continue)**
> >
> > To clearly explain our observations,  we will follow the notations in our revised paper (where $R^{\rm rob}_{\mathcal{D}}(\theta)$ is referred to as  $R^{\rm adv} _ {\mathcal{D}}(\theta)$ ) , while denoting $R _ {\mathcal{D}}(\theta):=\mathbb{E} _ {(x, y)\sim \mathcal{D}}\left[l _ {\theta}(x,y)\right]$ and $R _ {S}(\theta):=\frac{1}{m}\sum\limits _ {i=1}^{m}l _ {\theta}(x_i,y_i)$, representing the population and empirical risks of the model with parameter $\theta$ under the standard loss. Recall $\theta _ {t}$ denote the model parameter obtained from PGD-AT at iteration $t$.  In our IDE experiment IDE( $t$ ), a training set $\tilde{S} _ {t}$ is drawn from the perturbed distribution $\tilde{\cal D} _ t$,  and the model is trained from scratch by minimizing $R _ {\tilde{S} _ t}(\theta)$ using SGD, and denote the learned parameter by $\phi _ t$. Note that by the construction of
> > $\tilde{\cal D} _ {t}$ and $\tilde{S} _ {t}$, we have $R^{\rm adv} _ {\mathcal{D}}(\theta _ t) = R _ {\tilde{\cal D} _ {t}}(\theta _ t)$ and $R^{\rm adv} _ {S}(\theta _ t)= R _ {\tilde{S} _ {t}}(\theta _ t).$ Our experiments observed that the IDE testing error  $R _ {\tilde{\cal D} _ {t}}(\phi _ {t})$ correlates with the robust generalization gap $R _ {\tilde{\cal D} _ {t}}(\theta _ t) - R _ {\tilde{S} _ {t}}(\theta _ t)$. Note that there is no reason to believe that $\phi _ {t}$ is closely related to $\theta _ t$, particularly because of the complex min-max training dynamics in obtaining $\theta _ t$ at each iteration $t$ and because $\phi _ t$ is obtained by training from near-zero intialization (a starting point unrelated to $\theta _ t$). Thus this observed correlation, to the best of our knowledge, is novel and is not a re-discovery of some well-known effect of AT.
> >
> > This correlation suggests that the **training trajectory** of PGD-AT and the resulting robust overfitting is related to the increasing generalization difficulity of the induced distribution $\tilde{\cal D} _ t$ along PGD-AT iterations. This is the first key message of this paper. The second message of this paper is an explanation of this increasing generalization difficulty via a novel generalization bound, from which we discover that a property of the perturbation operator, namely, "local dispersion", plays a key role. This explanation is further corroborated by additional experiments (Figure 4).

---

> ### Author Response · Authors · 2023-11-22
> **Reply to major concern 2**
>
> We understand the reviewer's perspective that regards robust overfitting as the classical notion of overfitting under a different choice of loss function. However, classifical learning-theoretic bounds involve complexity measures (e.g. Rademarcher Complexity) that are difficult to characterize for deep neural networks and one must recourse to upper bounds of the complexity measures, making the bounds too loose to be relevant. This difficulty aggravates in AT when the loss function involves maximization over a norm ball, i.e., $\ell^{\epsilon}$.  Our approach unfolds the training dynamics, and enables us to obtain quantities (i.e. local dispersion) that closely track the model's generalization behaviour as it is updated along the training trajectory.
>
>  We agree that adversarial training is related the DRO literature, and we have included some of the DRO works in our section of related works in the revision. We must point out however that, to the best of our knowledge, all analyses of generalization in the DRO literature, including the ones the reviews pointed to, ignore the impact of training dynamics on the generalization behaviour of the learned models. As discussed earlier, such an approach often fails to be relevant to deep learning systems. The two papers (Kuhn et al. (2019) and Esfahani and Kuhn (2018)) pointed to by the reviewer are in fact perfect demonstration of the limitation of this approach.
>
> Specifically, Theorem 18 in "Wasserstein distributionally robust optimization: Theory and applications in machine learning" by Kuhn et al. (2019) and Theorem 3.4 in "Data-driven distributionally robust optimization using the Wasserstein metric: performance guarantees and tractable reformulations" by Mohajerin Esfahani and Kuhn, 2018) are much worse than the bound we provide in Theorem 1 of this paper. To see this, consider CIFAR 10 dataset, in which the input dimension is 3072. Acccording the bounds in Kuhn et al. (2019) and Esfahani and Kuhn (2018), high-probability generalization error decays at the rate of $m^{-1/3072}$, whereas in our bound, the error decays as $m^{-1/2}$, where $m$ is the size of the training sample (which was denoted by $N$ in their papers). In fact, it is precisely because these bounds are algorithm-agnostic, they insist that regardless of the loss function being the standard loss $\ell$ or the adversarial loss $\ell^{\epsilon}$ (as you defined), the generalization error decays at the same rate, hence unable to explain the drastic difference in generalization between AT and standard training.
>
> Nonetheless, thank you for pointing to the connection to the DRO literature. In light of this connection,  the approach developed in this paper may in fact translate to additional insights into the study of the DRO problem.

---

> ### Author Response · Authors · 2023-11-22
> **Reply to major concern 3**
>
> - In the abstract, the terms used will not be clear until one reads the paper. For example "evolution of the data distribution" will be defined later on and here it does not make meaning. Similarly "checkpoints", or "certain local parameters" are unclear yet. Also in the abstract, there is circular reasoning: initially, it is said this paper will explain why robust overfitting happens, but then "We observe that the obtained models become increasingly harder to generalize when robust overfitting occurs" -> Aren't we investigating the reason behind robust overfitting?
>
> We've considered your suggestion and completely rewritten the abstract.
>
> - "robust population risk $R_{\mathcal{D}}^{\text{rob}}$" is defined wrongly. This is adversarial risk. Robust refers to the model that is trained adversarially robust. The definition the authors provide is for any $\theta$, regardless of how they are trained. Overall, the paper uses "robust" and "adversarial" interchangeably, and this should be corrected.
>
> We've modified the "robust population risk" as "adversarial population risk", and denote it as $R^{\rm adv}_{\mathcal{D}}(\theta)$.
>
> - "As such, one may expect the generalization performance of a model trained on $\mathcal{D}_t$ is similar to that on $\mathcal{D}$" would contradict every paper that is concerned with distributional shifts and DRO.
>
> The sentence was not structured correctly, and is now removed.
>
> - Some terms are used before they are defined or abbreviated. Examples: "PGD" name is being used in the beginning before it is defined in S3.
>
> We've fixed this in our paper.
>
> - There is some mystery behind discussions. In the beginning, for example, overfitting is said to be "unexpected" -> why?
>
> In that sentence, we do not mean that overfitting is unexpected. What is meant is that much more severe overfitting in adversarial training compared to standard training is unexpected.
>
>
> - Some statements are informal. Examples: "A *great deal* of research effort has been spent" is not followed by any references. On the next page, "does robust overfitting have anything to do with this" is also informal.
>
> We have revised this paragraph
>
>
> - "In Dong et al. (2021a), the authors observe the existence of label noise in adversarial training" -> What is a label noise, could the authors please briefly clarify? That said, "in X et al. (...)" is the wrong format as "X et al." refers to authors rather than the paper -- maybe use `citep`? The paragraph later mentions "analysis of memorization" which is also not clear as we are talking about an algorithm. The sentence with "small loss" can also be revised.
>
> The label noise in Dong et al. (2021a) refers to that after adversarial perturbation, the original label may no longer reflect the semantics of the example perfectly. We have revised to paper to clarify this. We have fixed the citation formats and the mentioned phrases.
>
> - Page 1, "smoothing logits" -> not clear. In general, when a paper is cited, please make sure to not assume the reader knows the specific terminology.
>
> We have improved the writting here.
>
> - Section 2, after Figure 1 has "Having established generalization difficulty of the induced distribution". There are two issues here; (i) the sentence is really hard to grasp, what is the "generalization difficulty of the induced distribution"? (ii) after showing an experimental plot, the authors claimed something is established, but it is not. This pattern appears throughout the paper, plots or experiments are used as evidence for theoretical significance, but I find these not convincing. (Similarly, the second sentence of Section 5 starts with "This suggests".)
>
> - The paragraph at the end of page 2: Again, this is specific to the training method.
>
> We have revised this paragraph
>
> The experimental observation and the related conclusion in our work are all towards PGD-AT, this specific training method. We've added a few line of statements in our paper to clarify this point.
>
> - "The most popular adversarial training" is said to be PGD-AT, but this is specific to a class of learners, e.g., neural networks.
>
> We have revised the wording to restrict the context to neural networks.
>
> - Could the authors cite some generic paper for the PGD-AT as described from the end of page 3 onward?
>
> We have fixed this following your suggestion.
>
> - The described PGD-AT provides an approximate solution, not necessarily a strong solution. However, the conclusions drawn in this paper are for adversarial learning in general. This distinction could be clarified.
>
> We've clarified in the paper that our research focus on PGD-AT with deep neural networks instead of adversarial learning in general.
>
> - Section 6 is overall a heuristic approach, and informal. If the preceding sections were formal enough, this would be a good teaser, but at the current form, I think it makes the paper less formal.
>
> We have revised the section to improve its formality.

---

> ### Author Response · Authors · 2023-11-22
> **Reply to the questions**
>
> - Introduction: "error probability in the predicted label for adversarially perturbed instances" -> Is "probability" the correct word?
>
> We think "error probability" is an acceptable phrase in this context.
>
> - For clarity, can the authors specify what each "error" looks like, maybe via mathematical notation? In the above sentence, for example, "the error in the predicted label for adversarially perturbed instances" is mentioned, but the structure of adversarial attacks comes later. That said, I was also wondering, can the authors add a sentence discussing what part of their proofs would not work in the case of ℓp≠∞ norm attacks? Finally, the ℓp attacks, even when they are not restricted to ∞-norms, typically deserve a quick discussion on why such attacks are interesting.
>
> Should the reviewer insist that we insert a mathematical equation defining the error and restructure description in the introduction, we will do that in the final revision. But we think, for the introduction, it might be better to stay at a high level and defer the precise explanation to the body of the paper.
>
> - Page 2 says Dong et al and Yu et al. conflict 'to' each other (conflict "with" might perhaps be the better word): could the authors please explain a little bit about how these explanations vary? The sentence that follows says "each shown to" which probably is a typo.
>
> This is now fixed.
>
> - Before the PGD training is mentioned, the paper uses the phrase "along adversarial training". I think the "along" implies that the algorithm is iterative and again it is specific to the structure of the PGD-AT algorithm. Could the authors please mention this at the beginning of the paper? Similarly, after that "induces a new data distribution" is not immediately clear; I only understood what is meant here after I read the paper fully once. Please also clarify "at training step t" terminology.
>
> We have revised this part to improve clarity.
>
>
> - Right before Figure 1, could the authors please include the framework of IDE testing step by step? It does not need to be long but it would improve the readability. I understood the whole story much better in the first paragraph of Section 4.
>
> We have revised this part.
>
> - Can the authors please cite WRN-34 and PRN-18 when they first appear?
>
> This has been fixed.
>
> - End of Section 2: can you please elaborate on the "mismatch between theoretical settings and real-world practices"
>
> This remark is now deleted.
>
> - Could the authors please clarify what on page 4 "**distribution** of (v,y)=..." refers to? Is it the distribution whose randomness comes from ρ? Or do the authors mean the collection of training instances perturbed with respect to the underlying mechanism?
>
> We have added the phrase "random variable pair" before $(v, y)$ so that it is now clear that the distribution refers to their joint distribution.
>
>
>
> - I guess the name "checkpoints" is coined because it is expensive to do this IDE check at every iteration. Would it perhaps be less confusing if we do not mention checkpoints, rather say "IDE is recorded every k iterations"?
>
> We have now better explained what checkpoints are referred to.
>
>
> - The experiments are repeated only 5 times, isn't this also contradicting the stochastic optimization literature/curse of dimensionality etc., as the choice of ρ is crucial? Maybe the SAA literature has an answer to this. (Similarly the 10 pairs of ρ,ρ′ need discussion).
>
> Agreably, more repeatitions will improve our estimation. We will make an effort in that direction in the final revision.
>
> - Why are the parameters not CV'ed? It is ok to just cite a paper using similar combinations.
>
> Please clarify the meaning of CV and we will make an effort to revise.
>
> - Can Figure 3 (a) be plotted over the error rate range [0, 0.1]? It is hard to read on this scale.
>
> This choice of scale is for an easy visual comparison with Figure 3 (b).  The exact error values in 3(a) do not matter as much, since key message is to show its difference wtih 3(b).
>
> - Is there a reason why Section 5 uses f∈F notation rather than keeping the adversarial attacks explicit? Would the proof make sense if ε=0 as well? Also, the f∈F refers to **any** function, can the authors make it clear how the adversarial setting is used in the proof? I guess the only property of f about the adversarial training is the "boundedness of perturbation-smoothed loss) -- isn't this very loose?
>
>  We wish to note that the proof is not directly on adversarial training, rather it is proving a generalization bound for IDE experiments.
>
>
>
> - "measurable" -> With respect to which algebra? (I am asking because the norm space is explicitly defined).
>
> Here measurable is defined with respect to the standard Borel algebra.

---

### Official Review · Reviewer_jc1G · 2023-11-04

**Soundness:** 3 good
**Presentation:** 3 good
**Contribution:** 3 good
**Rating:** 8
**Confidence:** 4

**Summary:**

The paper investigates the phenomenon of robust overfitting in adversarial training. Through experiments conducted on models at various checkpoints along the adversarial training process, the paper shows that there is a correlation between the robust generalization gap and testing error on a dataset induced by adversarial perturbations on the model at that time (called the IDE testing error). Through experiments, the paper shows that the IDE testing error to a quantity termed expected "local dispersion" of the adversarial perturbations on the dataset at that time. The paper then presents a generalization bound on adversarial risk in terms of this expected local dispersion, thus supporting their earlier experimental finding. Overall, it is shown that the adversarial training induced distribution plays a key role in robust overfitting. The paper also conducts additional experiments on the adversarial perturbations at various checkpoints along the adversarial training process. It is shown that the perturbations increase in dispersion as the training progresses.

**Strengths:**

- Analyzing robust overfitting by conducting experiments on models at various checkpoints along the adversarial training process is a novel idea, to the best of my knowledge.
- The claim that adversarial training induced distribution plays a key role in robust overfitting is supported both via experiments, and via some theory.
- The "induced distribution experiments" offer interesting insights into the adversarial training process, and deserve further study.

**Weaknesses:**

- The results in section 6 on the dispersion of perturbations along the adversarial training process, although interesting, do not fit well into the main message of the paper. In fact, the results of experiments on the evolution of $d_\theta(x, y)$ seem to undercut the main message.

**Questions:**

- Why is $\rho$ in (5) taken to be from the uniform distribution on the hypercube as opposed to any other distribution centered at $x$ and supported on $B(x, \epsilon)$?
- How different is $Q_{x, y, \theta}$ from the perturbation $v$ in equation (5)? Can you please motivate why you study this "averaged" perturbation in place of directly studying $Q_{x, y, \theta}$?
- Using only 10 pairs of $(\rho, \rho')$ in the Monte-Carlo estimate of $\gamma_{B(x, \epsilon)}(Q_{x, y, \theta})$ seems awefully inadequate especially because the hypercube $B(x, \epsilon)$ is of large dimension. Typically, the number of samples you need to estimate an expectation over a hypercube increases exponentially with dimension. I have the same comment about the estimate of $d_\theta(x,y)$. What is the variance you observe for these estimates?
- Is there a reason for restricting the adversarial perturbations to $L_\infty$ norm only, as opposed to keeping it general to any norm?

---

> ### Author Response · Authors · 2023-11-22
> **Reply to your questions**
>
> Thank you very much for your careful review. Regarding the weakness that you pointed here, we will make an effort to revise section 6 to improve its presentation in the final revision. We were not able to do it during the rebuttal period, due to too much time spent on addressing a concern of another reviewer 3NwP.
>
> For your questions, we list our replies one by one over here:
>
> 1. We have reformulated the setting and now define the data distribution ${\cal D}$ as resulting from uniformly perturbing a more robust distribution ${\cal D}^{*}$. For this reason, $\rho$ drawn uniformly at random from the $\epsilon$-ball naturall arise. Please refer to Section 5 for our reformulation.
> 2. ${\mathcal Q}_{x, y, \theta}$ refers the perturbation operator, whereas $v$ refers to the output of the operator on input $x+\rho$.  There may be many perspectives to study this  perturbation operator. Here we choose to study its dispersive properties since this appears to be closely related to the generalization difficulty of distribution induced by it.
> 3. We have improved the experiments by sampling 250 pairs in the Monte-Carlo estimate. These experiments requires significant computing resources. In this new experiment, the variance of the estimate ranges from 0.0025 to 0.0166 at various checkpoints. (noting the estimate of ELD is about 0.43 to 0.6). Although this experiments can be improved by increasing the number of samples, we believe that the message from this experiments is already convicing. In the final revision, we will make an effort to further increase the number of samples in this estimation.
> 4. The main focus of this paper is to develop improved understanding of robust overfitting. Since in PGD-AT defined using other norms, robust overfitting does not arise as severely. This has been the reason that this paper is restricted only to infinity-norm based PGD-AT. Nonetheless, it may also be interesting to look into other kinds of perturbations to develop more general understanding of robust overfitting.

---

### Meta-Review · Area_Chair_jH81 · 2023-12-05

**Metareview:**

This paper shows that robust overfitting observed under adversarial learning is correlated with the uncertainty generated by the learning trajectory. This paper also derives an upper bound on the generalization error based on the claim. This problem is an important issue in adversarial learning. The authors' proposed idea is interesting, and several reviewers have acknowledged certain value in the paper's approach. However, the reviewers reached a consensus that the authors could not effectively refute several objections and counterexamples by the reviewers. It is also noted that the theoretical upper bound is somewhat inadequate to support the claims of this paper. The paper still needs to be improved to refute these points effectively.

**Justification For Why Not Higher Score:**

The weaknesses of this paper are clearly identified and there is consensus among the reviewers that there is room for improvement.

**Justification For Why Not Lower Score:**

N/A

---

### Decision · Program_Chairs · 2024-01-16

Reject